# Interpretable multimodal machine learning (IMML) framework reveals pathological signatures of distal sensorimotor polyneuropathy
Phong B. H. Nguyen[1,2,3], Daniel Garger[1,2], Diyuan Lu[1], Haifa Maalmi[3,4], Holger Prokisch [5,6], Barbara Thorand [3,7,8], Jerzy Adamski [9,10,11], Gabi Kastenmüller [1,12], Melanie Waldenberger [13], Christian Gieger [13], Annette Peters[7,8], Karsten Suhre[14], Gidon J. Bönhof[3,4,15], Wolfgang Rathmann[3,16], Michael Roden [3,4,15], Harald Grallert[7,13], Dan Ziegler[3,4,15], Christian Herder [3,4,15,18] ✉ & Michael P. Menden [1,2,3,17,18] ✉

## Abstract

**Background** Distal sensorimotor polyneuropathy (DSPN) is a common neurological disorder in elderly adults and people with obesity, prediabetes and diabetes and is associated with high morbidity and premature mortality. DSPN is a multifactorial disease and not fully understood yet.

**Methods** Here, we developed the Interpretable Multimodal Machine Learning (IMML) framework for predicting DSPN prevalence and incidence based on sparse multimodal data. Exploiting IMMLs interpretability further empowered biomarker identification. We leveraged the population-based KORA F4/FF4 cohort including 1091 participants and their deep multimodal characterisation, i.e. clinical data, genomics, methylomics, transcriptomics, proteomics, inflammatory proteins and metabolomics.

**Results** Clinical data alone is sufficient to stratify individuals with and without DSPN (AUROC = 0.752), whilst predicting DSPN incidence 6.5 ± 0.2 years later strongly benefits from clinical data complemented with two or more molecular modalities (improved ΔAUROC > 0.1, achieved AUROC of 0.714). Important and interpretable features of incident DSPN prediction include up-regulation of proinflammatory cytokines, down-regulation of SUMOylation pathway and essential fatty acids, thus yielding novel insights in the disease pathophysiology.

**Conclusions** These may become biomarkers for incident DSPN, guide prevention strategies and serve as proof of concept for the utility of IMML in studying complex diseases.

## Plain Language Summary

Distal sensorimotor polyneuropathy (DSPN) is a common neurological disorder in elderly adults and people with obesity, prediabetes, and diabetes in which there is tingling or numbness with or without pain. It is not fully understood why it develops. We developed a computational method that uses various sources of information to enable people with DSPN to be identified and also to predict which people might develop DSPN in the future. Further development of our method might provide additional information that can be used to prevent development of DSPN in people with obesity, prediabetes, and diabetes. Also, our method could potentially be adapted to enable other complex diseases to be better understood.

Type 2 diabetes (T2D) and its comorbidities have become a global challenge given the increasing case numbers and the enormous cost of diagnosis and treatments, putting burden on the public health management worldwide[1–4]. Distal sensorimotor polyneuropathy (DSPN) is the most common neurological complication in T2D which is characterised by a sensory loss of lower limbs, with or without neuropathic pain, caused by nerve damage[5]. Importantly, recent studies show that DSPN is also prevalent in elderly adults and people with prediabetes and obesity, thus affecting an increasing proportion of the general population[6,7].

DSPN diagnosis is challenging. It is based on evaluating the sensing ability of individuals, observing existing physiological conditions and morphological changes, and finally conducting neurophysiological measurements[7]. However, a large proportion of individuals with DSPN remain undiagnosed[8], and we lack computational methods to reliably

predict prevalent (i.e. cross-sectional) and incident (i.e. disease trajectory) DSPN. Furthermore, the complex pathogenesis of DSPN is not fully understood yet, and is anticipated to be multifactorial[9], attributed by the interplay of many intrinsic and extrinsic factors[7], thus rendering predictions challenging.

With the advent of multi-omics technologies, we are now able to conduct high-throughput assays that simultaneously characterise hundreds to millions of biomolecules across large patient cohorts[10–12]. As a result, the number of datasets with deep multi-omic characterisation has been exponentially increasing in recent years, e.g. the population-based KORA (Collaborative Research in the Region of Augsburg) F4/FF4 cohort which includes a subset of 1091 participants with DSPN label defined by the Michigan Neuropathy Screening Instrument (MNSI)[13]. Each participant in KORA is characterised with clinical data, genomics, methylomics, transcriptomics, proteomics, inflammatory proteins and metabolomics[13]. Rise of these large-scale multimodal datasets and computational integration frameworks are the prerequisite to gain insights in complex multifactorial diseases and comorbidities at multiple molecular levels[14,15], here exemplified with DSPN.

Statistical methods empower biological insights. For instance, to select genes associated with a certain phenotype, e.g. gene expression patterns in DSPN, the conventional method is setting a fixed significance threshold for a certain univariate statistical test and selecting genes that fall under the threshold[16,17]. While it is effective in identifying the most univariately significant genes, it tends to neglect smaller effect sizes which may cumulatively contribute to a multivariate model. In order to address this, gene set enrichment analysis (GSEA) is a powerful tool to prioritise functional relevant genes regardless of their global effect size, as it puts genes into context of biological signalling pathways using prior knowledge. Notably, GSEA generalises to other biomolecules such as proteins and metabolites, representing a potential approach to study complex systemic diseases[18].

There are several multimodality data integration strategies. The simplest approach is to concatenate all available features together before supervised learning[19,20]. This method is simple to implement, however, it requires extensive data processing and normalisation to incorporate heterogenous modalities encompassing vast amounts of features, thereby often neglecting important biological signals[20]. To address this, state-of-the-art integration methods such as ensemble stacking (meta learning) are employed to combine the powers of multiple data modalities and/or learning algorithms, whilst increasing weights of more predictive modalities/algorithms. This empowers to learn complicated structures and relationships of the data[21]. A critical assumption of multi-view learning, however, is that the single-view models should be independent[21]. This assumption is often violated in complex metabolic diseases, as there is a high level of redundancy and correlations amongst feature layers. Nevertheless, multi-view learning has proven to be superior compared to models leveraging concatenated feature space in crowd-sourced computational challenges[22,23].

In this study, we present the interpretable multimodal machine learning (IMML) framework, and exemplify its capability with DSPN classification and predicting DSPN onset over 6.5-years, i.e. prevalent and incidence predictions, respectively. IMML focuses on deriving predictive, interpretable and translational models leveraging sparse multimodal data. For this, we developed a two-step feature selection and integration machine learning framework. The first step extracted functionally relevant features of each molecular layer in isolation and leverages GSEA, whilst the second step benchmarked all combinations of data modalities based on cross-validated and regularised linear models. We hypothesised that well performing models at the minimum number of data modality would give insights into the disease aetiology of DSPN and its incidence, thus may improve diagnosis and pave the way for prevention strategies.

The derived framework successfully classifies cross-sectional DSPN and predicts future incident DSPN, as well as identify relevant and actionable biomarkers of the disease. In particular, the model achieves the AUROC of 0.752 and 0.714 for cross-sectional DSPN and incident DSPN,

respectively. Dissecting the model complexity shows that involving molecular data helps improving the prediction performance for incident DSPN, with $\Delta AUROC > 0.1$ compared to the clinical data-only model. Importantly, feature analysis shows multiple important signatures of incident DSPN such as up-regulation of inflammatory cytokines and down-regulation of SUMOylation process and essential fatty acids. These putative biomarkers serve as useful resources for future investigation to identify actionable biomarkers for interventions. These findings do not only help identifying individuals at risk of developing the disease but also shed light into the pathological mechanisms and important biomarkers that would help improve patients' life, further advancing precision medicine.

## Methods
### Population data
The population-based data in this study was obtained from the "Kooperative Gesundheitsforschung in der Region Augsburg/Cooperative Health Research in the Region of Augsburg" (KORA) platform[24,25]. Specifically, data from the KORA F4 (2006–2008) and the KORA FF4 (2013–2014) studies, both follow-up examinations of the population-based KORA S4 study (1999–2001), were used. All examinations were carried out in accordance with the Declaration of Helsinki, including written informed consent from all participants. The KORA study was approved by the ethics board of the Bavarian Chamber of Physicians (Munich, Germany). The data used in this study was obtained under a data sharing agreement with the Board of Management of KORA and all data owners. Initially there were 1,161 KORA F4 participants aged 62–81 years in the age group with the neuropathy examination module. We excluded 28 individuals with known type 1 diabetes, diabetes forms other than type 2, or unclear glucose tolerance status. In total, we leveraged 1133 individuals.

### DSPN assessment
We used the examination part of the Michigan Neuropathy Screening Instrument (MNSI) score to assess the status of DSPN for all participants of KORA F4 and KORA FF4, as described previously[25]. In the MNSI assessment, we evaluated the appearance of feet (normal or any abnormalities such as dry skin, calluses, infections, fissures, or other irregularities), foot ulceration, ankle reflexes and vibration perception threshold at the great toes which was assessed with the Rydel-Seiffer graduated C 64 Hz tuning fork[26]. The normal limits for vibration perception threshold, adjusted for age, were determined based on the method outlined by Martina et al. [27]. The MNSI score also included the bilateral examination of touch/pressure sensation using a 10-g monofilament (Neuropen)[28]. Therefore, the total MNSI score ranged from 0 (indicating normal in all aspects) to a maximum of 10 points. Considering the advanced age of the participants and the inclusion of the monofilament examination, we defined distal sensorimotor polyneuropathy (DSPN) as a score of equal or higher than 3 points[29]. Thus, participants with an MNSI score ≥ 3 in KORA F4 were considered as prevalent DSPN cases, whereas participants without DSPN in KORA F4 (MNSI < 3) but MNSI ≥ 3 in KORA FF4 were considered as incident cases. This definition meets the minimal diagnostic criteria for possible DSPN, as outlined by the Toronto Diabetic Neuropathy Expert Group[30].

Using this criterion, for prevalent DSPN analysis, among 1091 out of 1133 individuals having MNSI scoring records, there were 188 cases and 903 controls. For incident DSPN analysis, we only considered the 903 controls in the KORA F4 and examined their progression of DSPN status in the KORA FF4. Among these, we excluded 378 individuals that either did not participate or lacked MNSI scoring records in the KORA FF4. For the incident DSPN analysis, the remaining 521 participants were split into 131 DSPN cases and 394 controls. For both predictions of prevalent and incident DSPN, we only leveraged clinical and molecular features collected at the early time point of KORA F4.

### Data pre-processing
From the KORA F4 study we obtained six types of molecular data, including genomic (Affymetrix Axiom), transcriptomic (Illumina HumanHT 12v3

Expression BeadChip), proteomic (SOMAscan), metabolomic (Metabolon), methylomic (Illumina Methylation 450k) and a small panel of inflammatory proteins (OLINK) data, besides clinical records. Each molecular layer was standardised before downstream analysis by computing the *z*-score, which accounts for different distributions and numerical scales of features. Our analysis pipeline pre-processed the data in a modality-specific manner, as shown below.

## Processing of genomic data

Following microarray assay and initial imputation using the Haplotype Reference Consortium (HRC) as reference genome, the genomic dataset had 3788 samples and 7,545,537 SNPs. We used PLINK v1.07[31] for quality control of the genotype data. In particular, we removed SNPs that had equal or higher than 1% missing rate, less than 1% minor allele frequency (MAF) and significant deviation from Hardy Weinberg Equilibrium (HWE, $p < 1e-10$). We used the –annotate function in the MAGMA software[32] to annotate the SNPs to their associated genes, based on the gene location information from the human genome GRCh37, considering SNPs that locate 2 Mb upstream and 500b downstream of the genes. Following that we discarded SNPs that could not be annotated to a gene. We removed samples that had heterozygosity rates deviating more than three standard deviations from the mean across all samples. Finally, we filtered samples that had clinical records in the KORA F4 study. Eventually, the pre-processed genomic dataset included 1083 samples and 3,167,521 SNPs. We transformed the categorical SNP data into continuous alternative allele copy numbers (0, 1 or 2).

## Processing of transcriptomic data

The initial data generation, quality control and transformation were performed by the KORA study[33,34]. Specifically, the annotation of probes sequences to known transcripts was based on an annotation file provided by Illumina for HumanHT 12v3 BeadChip (using genome location of hg19). Only probes with the label "good" during mapping (probe sequence mapped uniquely to UCSC transcript) were included in this study. Furthermore, samples with less than 6000 detected probes were removed using Illumina's GenomeStudio. The data was log2 transformed and quantile-normalised using Bioconductor package *lumi*[35]. The samples were clustered using R and the outliers were removed. We obtained 993 samples and 48,804 transcripts for our analysis. The technical variables including amplification plate, RIN number and sample storage time were regressed out using the R package *limma*[17].

## Processing of proteomic data

The SOMAscan proteomic data was obtained from the KORA F4 study, including 1000 individuals and 1129 protein probes. One individual and 34 probes were removed due to low quality in accordance with the SomaLogic pipeline. Many probes mapped to multiple proteins/UniProt IDs so we transformed probe annotation into protein annotation. We also filtered for samples that had clinical records. In total, the dataset included 397 individuals and 1160 proteins.

## Processing of metabolomic data

The Metabolon metabolomic data obtained from the KORA F4 study included 1768 individuals and 525 metabolites, after initial quality control and transformation. Particularly, the data was $\log_{10}$ transformed and values that lied more than four standard deviations from the mean were set to missing. We additionally discarded metabolites that had more than 70% missing values. For the remaining metabolites we imputed missing values using k nearest neighbour algorithm. Furthermore, we discarded samples that had standardised Mahalanobis distance larger than four and samples that did not have clinical records. Finally, we leveraged 829 samples and 466 metabolites.

## Processing of methylomic data

The Illumina 450k Methylation M-value data was obtained from the KORA F4, which had already undergone filtering for detection rate and data normalisation. The original data had 1727 samples and 485,512 methylation probes. We leveraged the ChAMP pipeline for methylation data processing[36]. Specifically, we excluded probes spanning SNP regions and probes not associated to genes based on the Illumina annotation file. Then, we imputed missing data using k nearest neighbour algorithm. Finally, technical effects were regressed out using the ComBat function in the *sva* package[37]. Only samples with clinical records were included in this study. In total, we had 849 samples and 399,541 methylation probes for our analysis.

## Processing of inflammatory protein data

The OLINK inflammation panel included 92 inflammatory proteins which were measured in 1133 samples. We additionally removed 21 proteins due to low detection quality, as reported in our previous study[29]. In summary, we used 71 proteins for our analysis.

## Processing of clinical data

Clinical data obtained from the KORA F4 study included background information, diabetes and comorbidity status, lifestyle, blood biochemistry and medication usage for 1161 individuals[24,25]. Together with the filtering mentioned in the "Population data" section, there were 1133 samples remaining. Categorical variables were transformed using one hot encoding. Subsequently, variables having >10% missing values were discarded. In total, we leveraged 1133 individuals and 83 variables.

## Data partitioning for modality-specific feature selection

Each sample that was lacking at least one data modality was leveraged for modality-specific feature selection. For prevalent DSPN prediction included 710 genomic (141 cases and 569 controls), 621 transcriptomic (133 cases and 488 controls), 67 proteomic (9 cases and 58 controls), 476 metabolomic (76 cases and 400 controls), 495 methylomic (82 cases and 413 controls) and 720 clinical (142 cases and 578 controls) samples. The incident DSPN prediction leveraged 223 genomic (57 cases and 166 controls), 171 transcriptomic (47 cases and 124 controls), 58 proteomic (13 cases and 45 controls), 160 metabolomic (30 cases and 130 controls), 174 methylomic (38 cases and 136 controls) and 242 clinical (63 cases and 179 controls) samples. During gene set enrichment analysis, 100 stratified resampled splits were created for each of the modality-specific dataset, except proteomics due to limited sample size. We used 80% and 20% of samples for feature selection / training and testing, respectively.

## Data partitioning for final model training

Fully multi-modal characterised samples were used for final model training. For prevalent DSPN prediction, this was 285 samples (31 cases and 254 controls), whilst for incident DSPN prediction, it was 242 samples (54 cases and 188 controls). We created 100 stratified splits which leveraged 80% samples for feature integration/training, and the remaining 20% for model testing. We further partitioned the 80% training samples into stratified five folds for cross-validation. The cross-validation performance was used as a criterion for the FFS algorithm to select the optimal model. We never used any test data for neither model training nor tuning of model parameters.

## Gene set enrichment analysis

For the gene set enrichment analysis (GSEA), we leveraged the Bioconductor *fgsea* R-package[38], which is a more computationally efficient implementation compared to the original method[18]. For ranking genes, we used the *t*-statistics of the differential expression analysis from the Bioconductor *limma* R-package[17], which estimated the univariate association of the genes to the phenotype using a linear model. For calculating the enrichment score (ES), we used gene sets from the Reactome database[39]. Finally, the *p*-values were adjusted for multiple hypothesis testing with false discovery rate (FDR) < 20%, which is a lenient threshold allowing the selection of features with lower effect size, which may add predictive value in multivariate models in later integration steps.

The mapping of biomolecules to Reactome was customised for each data modality. For transcriptomic and proteomic data, we used the

Reactome gene set annotation with Entrez IDs. For metabolomic data, we used the Reactome metabolite set annotation based on ChEBI IDs.

For genomic data, we leveraged the MAGMA software[32] to estimate the gene effect and subsequently perform gene set analysis. First, we annotated SNPs according to nearby genes (2 kb upstream and 0.5 kb downstream), and consecutively used MAGMA to estimate the gene effect on the phenotype, taking into account the SNPs that were mapped to this gene. MAGMA estimated the gene effect by first conducting principal component analysis (PCA) using all SNPs linked to this gene, and afterwards used PCs to train a linear regression model predicting the phenotype. Finally, MAGMA computed the gene's p-value with F-test, and converted these to Z-values for the gene set analysis leveraging a linear regression model[32].

For methylomic data, we used the methylRRA method[40] to perform gene set enrichment analysis (GSEA) on the CpG probes. First, this required a differential expression analysis on the probes using the R package limma, followed by using the ranked list of p-values as input for methylRRA. To this end, methylRRA computed a p-value for each gene leveraging the ranking of all CpGs annotated to that gene by implementing Robust Rank Aggregation algorithm[41]. Consequently, the p-values were transformed into z-scores and were used for the GSEA to extract significant gene sets[40].

In all cases, we included the full set of Reactome signalling pathways at the lowest levels of pathway hierarchy to avoid redundancy, and at the time, ensure full unbiased coverage (Supplementary Data 2). Furthermore, except for the proteomic data, the GSEA was performed across the 100 stratified splits accounting for heterogeneity.

## Robust rank aggregation

We leveraged the implementation of Robust Rank Aggregation of Kolde et al.[41]. The molecules/molecule sets were ranked according to p-values, leading to a different ranked list per cross-validation/resampling run. Then, the rank distribution of each molecule/molecule set across all lists was tested against the random ranking distribution generated by permutation with the null hypothesis that there was no difference between the two distributions. The p-values of the test were adjusted for multiple hypothesis testing by multiplying the number of tested lists and additionally adjusted for the number of tested molecules/molecule sets by Benjamini-Hochberg method.

## Extraction of leading-edge genes

Leading-edge genes in upregulated gene sets are all genes from the beginning of the ranked gene list until the enrichment score (ES). In contrast, in case of down-regulated gene sets, leading-edge genes are from the ES to the end of the ranked gene list. Here, we leveraged 80% of all data for each of the 100 stratified resamples, did GSEA, extracted the leading edge molecules to train an elastic net model, and finally tested the model prediction on the left out remaining 20% of samples. For aggregating results of these 100 stratified resamples, we only considered predictive models (AUROC > 0.5), and leveraged a Robust Rank Aggregation (RRA) algorithm[41] with a false discovery rate (FDR) cutoff of 5%, which delivered a union of leading-edge gene sets. Afterwards, a GSEA was conducted on the union of leading-edge gene sets to extract the final consensus significant gene sets and leading edge molecules, which were subject to final model training.

## Clinical feature selection

For select clinical variables, we leveraged elastic nets using the R package caret with an 80% and 20% split for training and testing, respectively. We used weighted log-loss as the performance metric for hyper parameter tuning. The feature importance of the models was evaluated using the magnitude of t-statistics. Features with zero t-statistic were omitted. Finally, we used RRA with FDR cutoff of 5% to aggregate the important features across the 100 bootstraps.

## Iterative forward feature selection

The iterative forward feature selection (FFS) integrates multiple data modalities. It is based on 100 independent runs of fivefold cross-validation.

We tuned elastic net's hyperparameters alpha and lambda by grid search of 20 alphas and lambdas in range [0,1], resulting in 400 parameter sets. The chosen hyperparameter combination was the one having best mean performance across 5-fold cross-validations. In each run, we randomly sampled 80% of the dataset to perform five-fold cross-validation and the performance was tested with the remaining 20% data. For each fold of model training, elastic net models with weighted log-loss function to overcome class imbalances were implemented. Within the inner loop, a fivefold cross-validation selected the best data modality to add next. The prediction performance of the model was tested by predicting on the outer test set (20% samples). The prediction probabilities were calibrated using the Platt scaling method. In each step, the model adds the next best data modality based on increased performance until all data modalities are included.

## Extracting feature importance

We selected the optimal number of modalities based on their testing AUROC distribution. For this, the chosen number of modalities had a significant improvement in testing AUROC distribution compared to the previous number, and no significant improvement could be observed in the later complexities (Wilcoxon rank sum test, p < 0.05). After choosing the optimal number of modalities, the important features in the 100 models of that number of modalities were aggregated using RRA. Consecutively, we used the selected features to train a final elastic net model on the whole training dataset using the same settings as previous steps. The feature importance of the final model was accessed by t-statistics of model parameters.

## Benchmarking of feature selection and integration methods

For feature selection, we compared GSEA with the conventional thresholding methods. For feature integration, we benchmarked FFS, data concatenation and ensemble stacking approaches. Thus, in total there were six combinations of methods to compare. For the thresholding method, we implemented differential expression analysis using limma and selected features having p-values < 0.05.

Regarding feature integration, we benchmarked the FFS with ensemble stacking and feature concatenation. The latter concatenated all features into a single matrix before training the model. The ensemble stacking approach leveraged 100 independent runs with stratified resampling. This is, we generated 100 sets of stratified resamples, each consisting of 80% training and 20% test set (i.e. outer loop). Within each 80% training set, we further divided the data into 5-fold cross validation sets (i.e. inner loop). For each iteration of the inner loop, we trained an elastic net model on four out of five validation sets and made predictions on the remaining validation set. After five iterations, we obtained the predictions of all samples for that inner loop (corresponding to the 80% training set from the outer loop). We used this together with the ground truth (80% outer train) to train an elastic net meta model in the outer loop, and consequently tested the predictive performance on the remaining 20% test set. Importantly, the test sets were never used for any parameter optimisation nor training, and only leveraged for unbiased performance evaluation. This process was repeated for each data modality. For the feature analysis we implemented robust rank aggregation on the meta models of the ensemble stacking across 100 resamplings, then extracted the individual feature importance.

## Statistics and reproducibility

The multiomic datasets were preprocessed by the KORA study using customised software mentioned above. The development of the computational framework and the statistical analyses were conducted using the R packages and independent software detailed above. To reproduced the analysis results, one can obtain the data from https://www.helmholtz-munich.de/en/epi/cohort/kora/kora-studienzentrum, following a data sharing agreement with the KORA study. Details about sample sizes, types of data and code availability could be found in "Methods", "Data availability" and "Code availability".

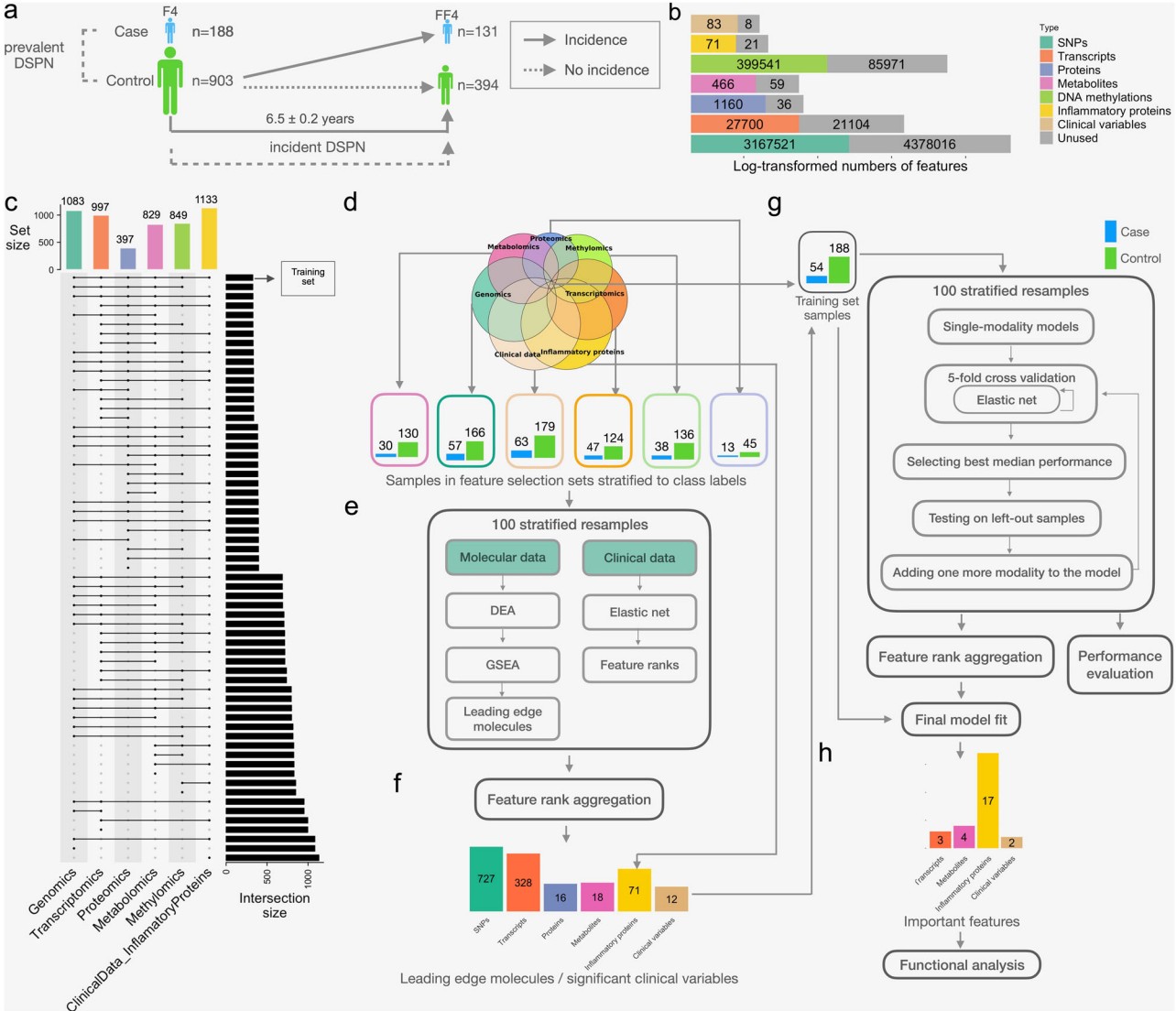

**Fig. 1 | Workflow of interpretable multimodal framework for feature prioritisation, DSPN classification and disease incidence prediction. a** Distribution of samples across time points (KORA F4 and FF4), disease status (case or control) at baseline (KORA F4) and follow-up (KORA FF4) and prediction tasks. Both models were trained on the same set of F4 features but different labels and a subset of samples. **b** Number of features stratified according to data modalities. In grey are removed features after pre-processing. **c** Number of samples characterised within each data modality and their overlaps in KORA F4. **d** Fully characterised samples in KORA F4 were exclusively leveraged for **g** the second and final training step, whilst the remaining sparse samples were used for **e** prior feature prioritisation: All molecular features were shortlisted based on differential expression analysis (DEA),

gene set enrichment analysis (GSEA) and their leading-edge genes ("Methods"), whilst clinical features were ranked according to feature importance of elastic net models. **f** Features for the final training step were selected based on rank aggregation ("Methods"). **g** The final training set contained 54 DSPN cases and 188 controls in KORA F4. In the second step, elastic net models determined the optimal number of modalities, features and combination of modalities. These models implemented forward feature selection in a nested cross-validation, using weighted log loss to account for class imbalance, and finally 100 stratified resampling during training and rank aggregation ("Methods"), thus returning **h** the refined and final model further subject to functional analysis for gaining insights in DSPN pathophysiology.

## Reporting summary

Further information on research design is available in the Nature Portfolio Reporting Summary linked to this article.

## Results

We leveraged the population-based KORA study including participants aged 62–81 years with clinical examination of DSPN from the F4 (2006–2008) and FF4 (2013–2014) surveys ("Methods"). The earlier F4 time point surveyed 1091 individuals of whom 622 were followed up at the later FF4 time point. We used the established Michigan Neuropathy Screening Instrument (MNSI) to assess and define the DSPN status as described in previous studies[25,29]. Using MNSI, we identified 188 DSPN cases and 903 controls at F4, and 131 controls who developed DSPN between F4 and FF4

(Supp. Tables S1 and S2). The first machine learning task was to predict DSPN prevalence at F4 (Fig. 1a). The second task was to predict whether controls at F4 will develop incident DSPN during the period from F4 to FF4 (Fig. 1a).

Data modalities included in this study were genomics, transcriptomics, proteomics, metabolomics, methylomics, clinical attributes and a panel of inflammatory proteins. The modalities vary greatly in number of features, ranging from 91 clinical attributes to >7.5 million single nucleotide polymorphisms (SNPs; Fig. 1b). After data type-specific processing ("Methods"), the number of features was drastically reduced, e.g., only approximately 42% of the assayed SNPs were used for subsequent analyses ("Methods"; Fig. 1b). Participants in KORA were sparsely characterised with varying overlaps of data modalities (Fig. 1c).

**The two-step feature selection and integration machine learning framework**

The IMML framework is based on a two-step approach: (i) Extensive feature engineering and selection process (Fig. 1d–f; Fig. S1; "Methods"), and (ii) final model training (Fig. 1g, h). For enriching biological signals and reducing feature space, we used 849 samples which were lacking at least one data modality, whilst the remaining completely characterised 242 samples were exclusively used for the final model training. Both subsets of data for feature selection and model training and testing were subject to PCA analysis using clinical information to ensure was no potential bias in sample selection (Supp. Fig. S2).

For the first step, i.e. feature engineering and selection processes, we prioritised predictive and biologically relevant biomolecules regardless of their effect sizes ("Methods"). We observed that GSEA-based methods significantly outperformed threshold-based methods (Wilcoxon Rank Sum test, $p$-value = 3.758e−12; Supp. Fig. S3). Therefore, we implemented differential expression analysis (DEA)[17] followed by GSEA[18] to extract a list of molecule sets corresponding to signalling pathways that may be pivotal in DSPN development. This process was repeatedly performed to account for variability ("Methods")[41]. Finally, we extracted the leading-edge molecules, i.e. those that drive the enrichment of molecule sets[18]. We obtained between zero and 25 significantly enriched molecule sets per data modality (Supp. Fig. S4 and S6; Supp. Table S3, Supplementary Data 1), from which we extracted up to 727 leading-edge features (Supp. Figs. S1, S5 and S7). In addition, for clinical feature selection, we trained elastic net models and leveraged rank aggregation to retrieve 13 predictive clinical features ("Methods"; Fig. 1e, f; Suppl. Fig. S1).

For the second step, i.e. final model training and multimodal data integration, we leveraged the short-listed features from the analyses above. The final model was trained with an embedded feature selection whilst balancing number of modalities. We benchmarked three feature integration methods, i.e. forward feature selection (FFS), ensemble and concatenation of all features together (Suppl. Fig. S3a; "Methods"), and observed best performance with GSEA-FFS followed by GSEA-ensemble stacking (Suppl. Fig. S3b). When comparing the performance of the FFS and ensemble stacking methods using all modalities and with GSEA as the feature selection approach, the FFS algorithm achieved marginally higher predictive performance (Suppl. Fig. S8a). Both methods retained inflammatory proteins as the most predictive features, however, the GSEA-FFS was further able to detect clinically relevant signals from other modalities (Suppl. Fig. S8b). Therefore, we implemented an iterative FFS algorithm with resampled cross-validation ("Methods").

To select the machine learning algorithm for DSPN prediction, we compared the predictive performance of elastic net, random forest and support vector machine, the latter leveraged linear and radial kernels (Suppl. Fig. S9). For this we performed 100 matched resamples with forward feature selection. Elastic net outperformed the other three machine learning algorithms in both prevalent DSPN (Suppl. Fig. S9a–d) and incident DSPN predictions (Suppl. Fig. S9e–h). Best performances in prevalent DSPN (AUROCs of 0.737) and incident DSPN (AUROCs of 0.708) predictions were observed at 1-modality and 3-modality models, respectively. Notably, none of the other machine learning algorithms reached AUROC higher than 0.700 at any number of modalities.

For each iteration of resampling, the most predictive combination of modalities were selected based on cross-validation (Fig. 1g, h; Suppl. Fig. S1). Analysis of the final model returned predictive modality combinations, which became subject to functional analysis for DSPN classification and incidence prediction in the following sections.

**Clinical data can sufficiently stratify individuals with and without DSPN**

In a clinical setting, all suspected DSPN patients are thoroughly clinically characterised and neurologically evaluated. Therefore, our FFS algorithm used KORA clinical attributes as baseline input, and consecutively, evaluated the gained performance by adding more molecular modalities to classify

DSPN (Fig. 2a). Metabolite and protein features were the most frequently added across 100 iterations, while transcripts were usually added last (Fig. 2a). However, the baseline clinical model significantly outperformed any more complex model (Wilcoxon rank sum test, $p < 2.22e−16$; Fig. 2b, c; Suppl. Figs. S10a and S11). The clinical model had a median area under the receiver operating curve (AUROC; "Methods") value of 0.752 with an interquartile range (IQR) of 0.686–0.821 and 95% confidence interval (CI) of 0.733–0.770, whilst the best performing model with molecular data only achieved a median AUROC of 0.583 with IQR of 0.539–0.627. This suggested that clinical variables alone are sufficient to stratify individuals with and without DSPN.

To further dissect the predictive component of clinical attributes, we extracted the most important clinical features from 100 resampled and cross-validated models. For this, we leveraged robust rank aggregation (RRA; FDR < 5%), and used these within the final model ("Methods"). After computing $t$-statistics of model parameters, four variables had non-zero $t$-statistics, including age, waist circumference, height and whether the patient had neurological illnesses (self-reported during interview; Fig. 2d). The principal component analysis (PCA) of these four clinical variables empowered the segregations of cases and controls (Fig. 2e). When we further stratified the prediction probabilities to individual samples and ranked them according to mean probability, most cases were ranked higher than controls, although there were a few outliers (Fig. 2f). Values of age, waist circumference and height were significantly higher in cases compared to controls ($p < 0.05$, Wilcoxon Rank Sum test) while having neurological illnesses was significantly enriched in DSPN cases ($p < 0.05$, Fisher's Exact test; Fig. 2g; Suppl. Fig. S12).

**Molecular data improves DSPN incidence prediction**

DSPN incidence prediction was strongly enhanced by integrating clinical and molecular data. In contrast to clinical baseline models (Suppl. Fig. S13a, b), we observed a strong benefit in leveraging molecular modalities for predicting whether participants of the KORA F4 cohort will develop DSPN or not within the next $6.5 \pm 0.2$ years (Fig. 3a, b; Suppl. Fig. S13c). The baseline DSPN incidence model achieved a median AUROC of 0.603 with an IQR of 0.543–0.676 and 95% CI of 0.588–0.624. This was significantly outperformed by adding either one or two additional molecular data modalities (Fig. 3a, b; Suppl. Figs. S10b and S14), i.e. median AUROC of 0.678 with an IQR of 0.612–0.752 and 95% CI of 0.652–0.692 and AUROC of 0.700 with IQR of 0.651–0.774 and 95% CI of 0.686–0.722, respectively (Wilcoxon rank sum test, $p = 1.9e−16$ and $p = 2.9e−11$, respectively). In essence, molecular features significantly enhanced DSPN incidence prediction. We observed that inflammatory proteins were >80% the first picked molecular layer, followed by metabolites, whilst SNPs seemed to carry the least predictive information (Fig. 3a). The performance tended to saturate at 3-modality models as adding more modalities did not significantly improve the performance anymore (Wilcoxon rank sum test, $p = 0.95$), i.e. 4-modality models had a median AUROC of 0.714 with an IQR of 0.640–0.774 and 95% CI of 0.684–0.720 (Fig. 3b). We observed similar results and a saturation of performance at 3-modalities, when not enforcing clinical attributes as baseline modality, which were still selected in 57% of all 3-modality models (Suppl. Fig. S15). In essence, prediction of DSPN incidence strongly benefited from adding two or more molecular modalities, and saturated at the 3-modality models.

For feature importance analysis of the final model, we selected 3-modality models ("Methods"). The prediction probabilities of incident cases were significantly higher compared to controls (Wilcoxon rank sum test, $p < 2e-16$; Fig. 3c). We obtained 26 features with non-zero $t$-statistics including 17 inflammatory proteins, four metabolites, three transcripts and two clinical variables (Fig. 3d; "Methods"). Most of the predictive power stemmed from inflammatory proteins, whilst two transcripts (CDC42 and SP3) and two metabolites (caprate and linolenate) displayed the largest $t$-statistic magnitude (Suppl. Fig. S16). PCA analysis on these 26 features illustrated that they enable the prediction of DSPN incidence (Fig. 3e).

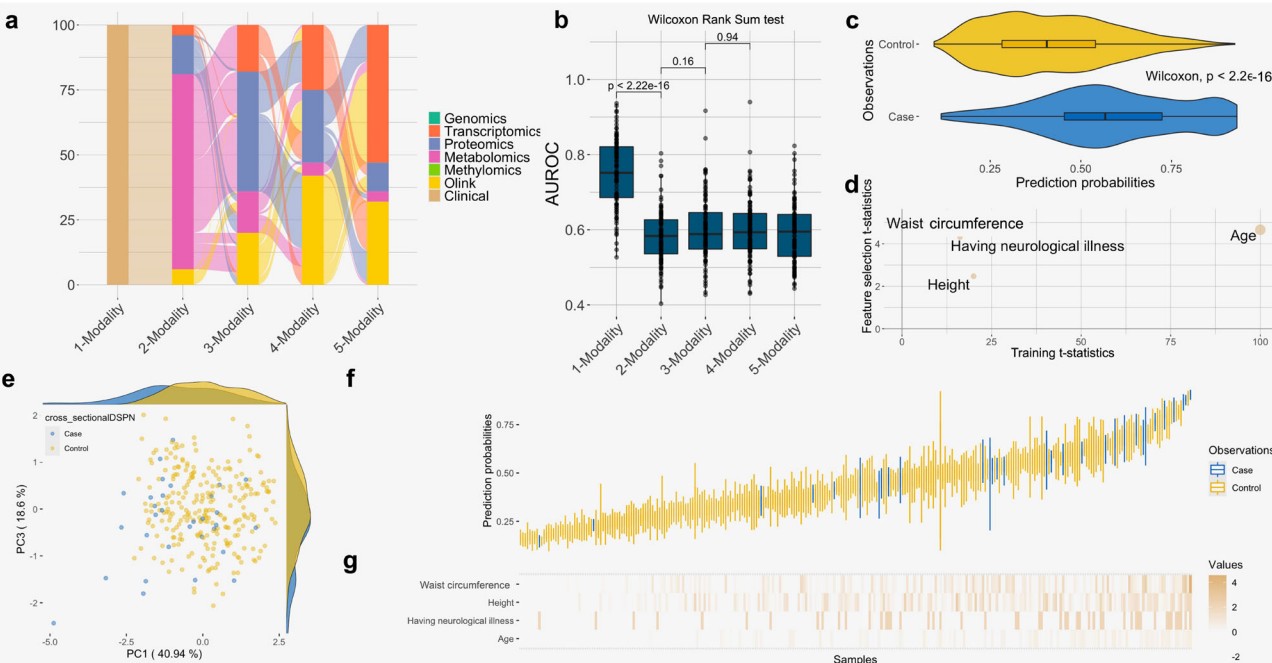

**Fig. 2 | The clinical model can sufficiently stratify DSPN prevalence.**
**a** Classification of DSPN first leverages clinical attributes, and cumulatively adds molecular modalities with forward feature selection ("Methods"). Here shown for 100 cross-validated models. **b** Test set performance of DSPN classification leveraging between one to seven data modalities. Error bars of the boxplot indicate 95% CI. **c** Prediction probabilities of samples in the 100 left-out test sets leveraging clinical features only, stratified into true labels (case and control). **d** Feature importance of

the final model based on clinical attributes alone applied to training and feature selection set ("Methods"). **e** PCA leveraging the four most important clinical features shown in panel **d** to stratify cases from control. **f** Distribution of the test prediction probability of all samples of 100 resampled and cross-validated models.
**g** Normalised values of the four most important clinical features. The order of samples corresponds to panel (**f**).

When stratifying the prediction probabilities of the 3-modality models into individual samples and ranking them based on their means, most of the incident DSPN cases were concentrated in higher probability regions, in contrast to controls (Fig. 3f). Furthermore, higher prediction probability also corresponded to higher concentrations of many model-important inflammatory proteins and lower concentrations of caprate (Fig. 3g). Incident DSPN cases were significantly enriched for low physical activity (Fisher's exact test, $p = 0.003$; Suppl. Fig. S17). In essence, levels of inflammatory proteins and metabolites were significantly different between people transitioning to DSPN compared to those who did not (Wilcoxon rank sum tests, $p < 0.05$; Suppl. Fig. S17), highlighting the important role of molecular features to predict DSPN incidence.

### Increased inflammation, reduced levels of SUMOylation and essential fatty acids as important signatures of incident DSPN

For gaining further insights into the prediction of incident DSPN, we investigated predictive features in the context of the initial GSEA-based feature selection. To this end, we created a network of features connecting all molecular layers by shared signalling pathways ("Methods"; Fig. 4a; Suppl. Fig. S7). For this, all biomolecules were connected to any other leading-edge molecule according to Reactome[39]. We identified two large subnetworks of 15 predictive features containing nine inflammatory proteins, three transcripts (CD42, SP3 and ITSN1) and three metabolites (caprate, linolenate and adrenate; Fig. 4a).

Inflammation is an important signature of incident DSPN prediction, which is evident by the increased frequency of functional important inflammatory proteins in the identified large subnetwork (Fig. 4a). To gain further understanding of their role, we performed GSEA on the proteomic training data focusing on gene sets involving inflammation. Binding of chemokines to their receptors was significantly upregulated (Fig. 4b; adjusted $p$-value = 0.008), as well as signalling of G protein-coupled receptors (GPCR; Suppl. Fig. S18; adjusted $p$-values < 0.2).

Transcriptomic modality encompassed consistently significant gene sets. In particular, downregulation of SUMOylation-related signalling pathways were consistently observed in both feature selection and training sets (adjusted $p$-values < 0.2; Fig. 4c; Suppl. Fig. S18). These included SUMOylation of proteins involved in DNA replication and DNA damage response and repair. In addition, the gene set involved in transport of mature RNA from nucleus to cytoplasm was significantly downregulated (Fig. 4c; Suppl. Fig. S18).

Interestingly, all metabolites in the subnetwork were fatty acids (Fig. 4a) and all were significantly downregulated, i.e. caprate, linolenate and adrenate. As a result, GPCR pathways related to fatty acids activity were significantly downregulated (Fig. 4d; Suppl. Fig. S18; adjusted $p$-values < 0.2). Other significant metabolomic pathways included the downregulation of fatty acid-related signalling and synthesis, secretion and inactivation of glucagon-like peptide-1 (GLP-1; Suppl. Fig. S18), and upregulation of transport of organic anions (Suppl. Fig. S18).

Overall, functional analysis of the predictive features revealed molecular signatures of incident DSPN. Particularly, the up-regulation of several inflammatory proteins and downregulation of SUMOylation-related transcripts and essential fatty acids are the most significant patterns.

### Discussion
DSPN is a complex disease attributed to multiple and heterogeneous risk factors[7]. Thus, integration of sparse multimodal data is a prerequisite for a deeper understanding of the disease pathophysiology. In order to address this, here we present the IMML framework, which allows prediction of prevalent and incident DSPN status based on clinical and molecular characterisation. We achieved good performance for both prediction tasks, i.e. AUROC > 0.7. Furthermore, the IMML two-step approach empowered the analysis of sparse clinical and molecular data, which is common in biomedical research. Utilising the modality-specific non-overlapping samples for feature selection increased the number of accessible samples and reproducibility of molecular patterns across different datasets.

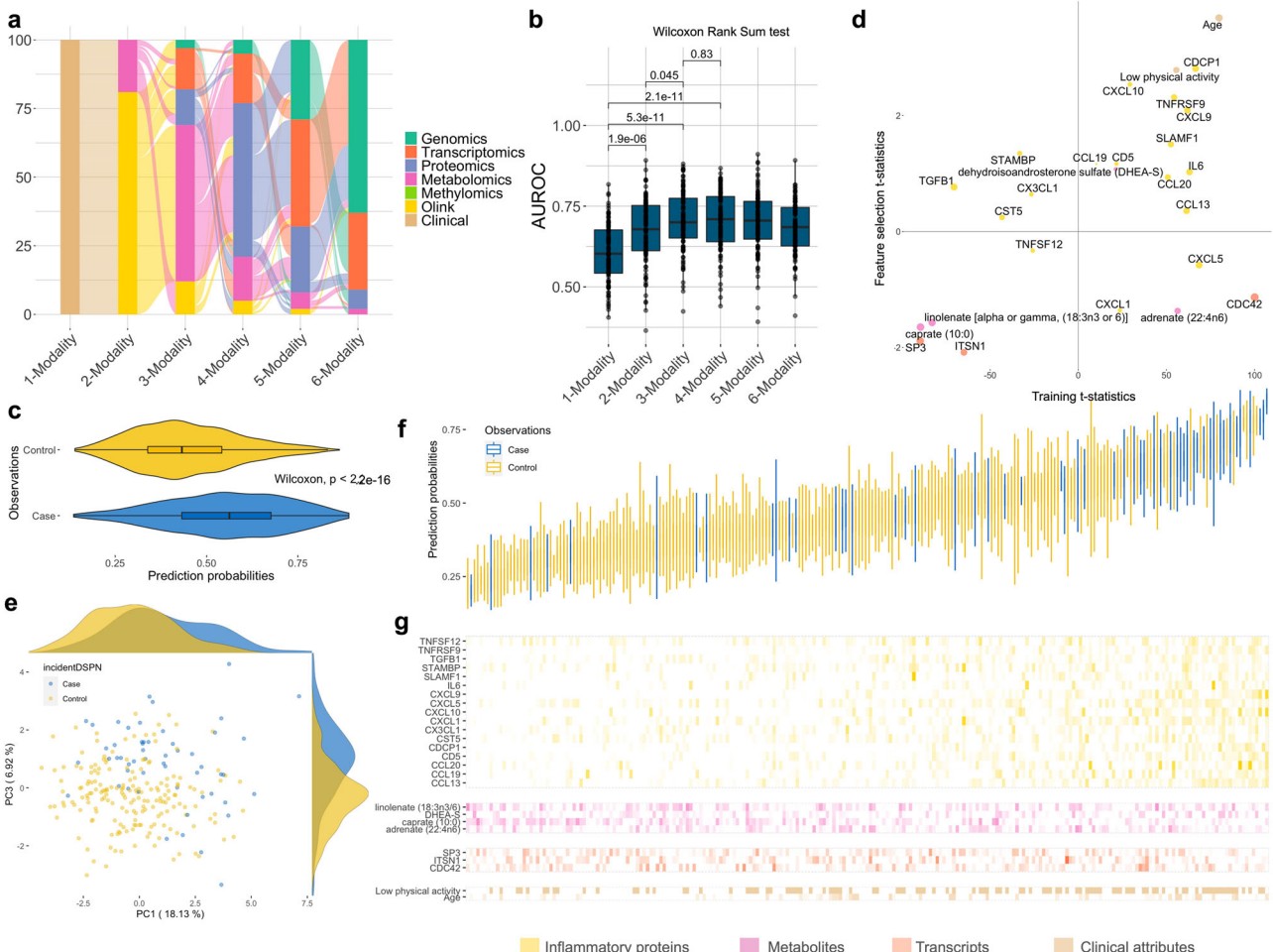

**Fig. 3 | Predicting DSPN incidence benefits from molecular data. a** Each model starts with clinical attributes at baseline, and consecutively increases the number of modalities by adding the next molecular modality with feed forward selection for 100 cross-validated models ("Methods"). **b** Performance of all model complexities to predict patient trajectories. Error bars of the boxplot indicate 95% CI. **c** Prediction probabilities of samples in the 100 left-out testing sets using the optimal mode of the corresponding iterations, stratified into true labels (case and control). **d** Important features of the final model. *x*-axis represents the signed model important scores (*t*-statistics) of the features in the training set, *y*-axis represents their t-statistics in the feature selection set. **e** PCA leveraging the most important features of the final model in panel (**d**). **f** Waterfall plot of prediction probability of all samples across 100 resampling steps. **g** Normalised values of the important features in panel (**d**) stratified by individual samples and ordered according to panel (**f**). Features belonging to the same data modality are grouped together.

The analysis of prevalent DSPN (classification of case-control in the F4 population) suggested that the clinical model (using only clinical variables) outperformed the concatenated models (using clinical + molecular variables) in prediction. Then, feature analysis of the clinical model suggested that age, height, neurological illness and waist circumference were the most important factors that influence the prediction of prevalent DSPN. Age and height have been reported to be associated with prevalent DSPN[42]. The neurological comorbidity status of patients is not used to classify DSPN yet, however, there might be an intrinsic neurological mechanism that links DSPN to other neurological illnesses. Finally, waist circumference is strongly correlated with BMI, which has been reported to be a risk factor for developing DSPN[43]. From a clinical perspective it is worth mentioning that only waist circumference represents a modifiable risk factor which emphasises the role of obesity prevention and treatment also in the context of DSPN. In summary, for prevalent DSPN, our analysis is confirmatory of previous studies with respect to these clinical variables. However, here we report the clinical variables in the context of a comprehensive multi-modality analysis of DSPN prevalence, thus adding another layer of information to the model.

In the case of incident DSPN prediction, the molecular variables added prediction value as they helped improve the prediction performance (higher AUROC values) compared to the clinical model alone. Feature analysis

detected multiple important and potentially actionable biomarkers such as inflammatory proteins, SUMOylation-related transcripts and essential fatty acids. Although the association between inflammatory proteins and incident DSPN has been reported before[25,29], there are as yet no data from population-based studies such as ours implicating SUMOylation-related transcripts and essential fatty acids in the development of DSPN so that these findings are novel and merit further investigation in other cohorts. Additionally, none of these biomarkers and pathways has been reported before in the context of our novel multi-modality analysis of DSPN incidence.

Feature analysis suggested the crucial role of subclinical inflammation in the development of DSPN. We found that 18 out of the 27 most important incident DSPN features were inflammatory proteins. Our finding was consistent with previous studies showing the predictive value of pro-inflammatory cytokines in DSPN[25,29]. One cytokine (IL-6), five chemokines (CXCL9, CXCL10, CCL13, CCL19 and CCL20) and five soluble forms of transmembrane proteins (CDCP1, SLAMF1, TNFRSF9, TNFRSF11B and CD5) were upregulated at baseline in patients with incident DSPN, suggesting a proinflammatory process which could be observed before DSPN onset[25,29]. CXCL9 and CXCL10 have been shown to directly impact neurotoxic effects[29]. In addition, nerve-derived chemokines may play a role in attracting immune cells to further damage stressed neurons[29]. In accordance

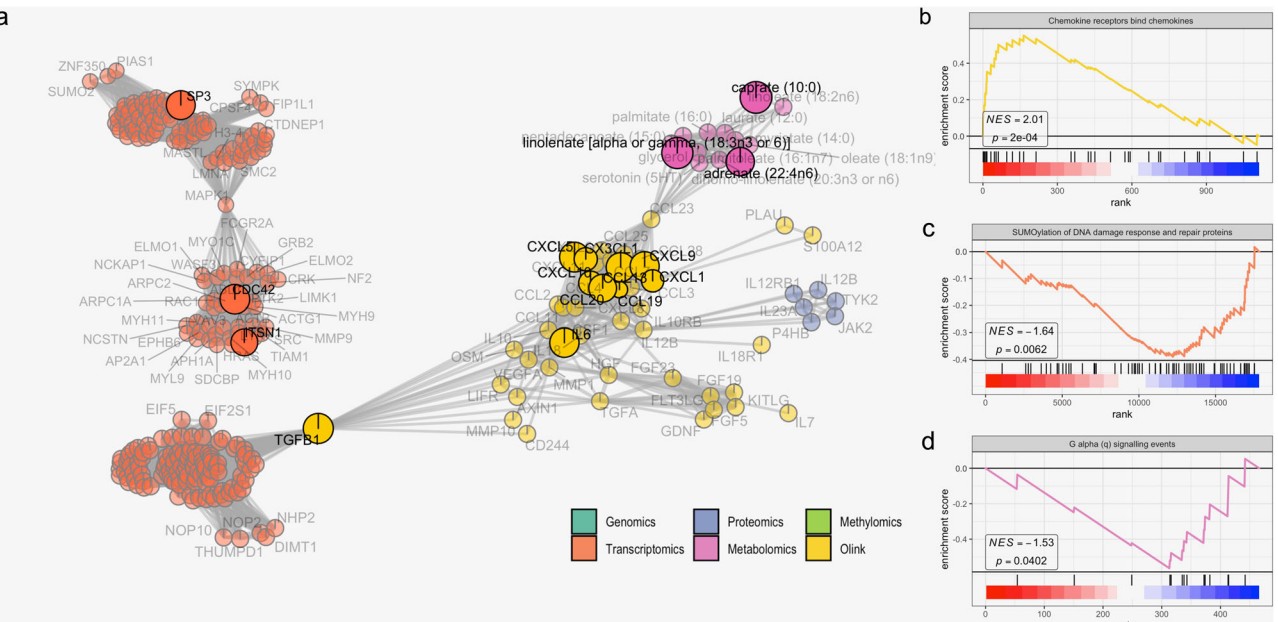

**Fig. 4 | Enrichment of inflammatory cytokines- and essential fatty acids-related pathways as important signatures of DSPN progression. a** Sub-network of important features to predict development of DSPN. Each node is a feature coloured according to its data modality. Edges are the number of shared molecule sets between two nodes. The important features in the final model are highlighted and labelled in black. Below are examples of enriched molecule sets associated with **b** inflammation-related proteins, **c** transcripts and **d** metabolites: **b** The upregulation of "Chemokine receptors bind chemokine" gene set. **c** SUMOylation of DNA replication proteins. **d** G alpha (q) signalling events. Molecules are ranked in decreasing order of *t*-statistics, with ticks representing molecules that belong to the examined molecule set.

with the upregulation of these proteins, signalling pathways downstream of GPCR signalling, specifically involving chemokine-induced inflammation, were also significantly up-regulated. Subclinical inflammation is an established hallmark of DSPN, as people affected by the disease often have elevated levels of pro-inflammatory cytokines that are associated with nerve damage[25,29,44,45]. It has been hypothesised that a cross-talk of innate and adaptive immune cells contributes to DSPN[29], which is further supported by our study, but requires further mechanistic validation.

Remarkably, inflammatory effects were observed in the blood samples prior to disease onset. Thus, the predictive pro-inflammatory cytokines, chemokines and transmembrane proteins observed in this study could represent modifiable risk factors and therefore therapeutic targets for disease prevention. For example, salicylate was reported in many studies to have inhibitory effects on production of cytokines and chemokines[46]. In addition, novel treatment approaches targeting IL-1beta-related mechanisms have been demonstrated to reduce subclinical inflammation and have beneficial effects on cardiometabolic risk[47,48], and may be generalisable for DSPN. Beyond pharmacological approaches to attenuate subclinical inflammation, it is important to emphasise that subclinical inflammation is triggered by a range of other modifiable risk factors such as high-calorie diet, certain nutrients, physical inactivity, obesity, psychosocial stress and sleep disturbances so that lifestyle changes represent another option for intervention[49].

The transcriptomic layer also gained attention as one of the most important predictors of DSPN. Particularly, significant down-regulation of the small ubiquitin-related modifier (SUMO) pathway was consistent with a recent study[50], which demonstrated SUMO posttranslational modifications are involved in glycolysis. Furthermore, the tricarboxylic acid (TCA) cycle plays a crucial role in maintaining important metabolic processes in sensory neurons, and deficiency of SUMO activity causes damaging effects which may specifically contribute to DSPN pathogenesis[50]. Although this enrichment analysis has to be interpreted with caution due to the small sample size, it is noteworthy that oxidative stress and inflammation have been proposed as mediators linking hyperglycaemia and impaired SUMOylation in diabetic polyneuropathy and that aberrant SUMOylation has also been

implicated in the aetiology of neurodegenerative diseases[51]. Thus, this finding extends the aforementioned results on inflammation, corroborates other studies and may point towards another mechanism how DSPN risk could be targeted by addressing modifiable risk factors leading to inflammation and oxidative stress.

Three fatty acids were identified as potential biomarkers of incident DSPN, i.e. caprate, linolenate and adrenate. Capric acid, also known as decanoid acid, binds to the α-amino-3-hydroxy-5-methyl-4-isoxazolepropionic acid (AMPA) receptor, a glutamate receptor that mediates synaptic transmission in the brain. Capric acid has antioxidative effects in neuronal cells[52] and has been implicated in the amelioration of several neurological diseases[53] so that further studies of the potential role of decreased capric acid levels for the development of DSPN appear promising. Adrenic acid-derived epoxy fatty acids have anti-nociceptive properties and can reduce inflammatory pain[54] so that a link between lower levels and higher risk of DSPN appears biologically plausible. Linolenic acid and adrenic acid, or all-cis-7,10,13,16-docosatetraenoic acid, are also essential polyunsaturated fatty acids (PUFA), which are precursors of more potent derivatives such as arachidonic (omega-6, ARA) and docosahexaenoic (omega-3, DHA) acids, which serve as either building blocks of cell membrane or substrates for the synthesis of inflammation-related compounds and are involved in neural development processes[55,56]. Furthermore, DHA reduces pro-inflammatory cytokines and induces anti-inflammation cytokines, which is consistent with the observed patterns of inflammatory proteins in our datasets[55]. In addition, certain groups of GPCR called free fatty acid receptors (FFAR), such as GPR40/FFAR1 and GPR120/FFAR4, are activated by PUFAs and medium-chain fatty acids (MCFA), such as capric acid, to regulate many cellular processes, i.e. insulin secretion, inflammation, neural cognitive and sensory function[57]. Overall, caprate, linolenate and adrenate have not been linked to DSPN in detailed investigations but nevertheless highlight the possibility that they should be modifiable risk factors that could be modulated by specific dietary interventions or dietary supplements. Importantly, experimental results suggested that PUFA might be a potential agent to treat DSPN[58,59], subject to future studies focussing on high-risk individuals assessing the

potential preventive and therapeutic properties of dietary fatty acids in this context.

One strength of our study's design is the utilisation of population-based prospective data from a large cohort (KORA F4). The KORA cohort contains repeated assessment of DSPN status using identical examination methods at two timepoints, which allows studying both prevalent and incident DSPN. The fact that the mean follow-up time was 6.5 years and that we do not have data on DSPN diagnosis between both studies means that our data cannot be extrapolated to considerably shorter or longer time-periods than 6.5 years. It is possible that different variables may be more powerful for short-term or very long-term prediction of DSPN which needs to be addressed in future studies. Furthermore, we presented an innovative machine learning framework to model incidence of DSPN by integrating multi-omic and clinical data. Previous efforts either focused on classifying the disease in a cross-sectional context, lacked multi-omics integration strategies or exhibited limitations of univariate statistical analyses[29,60–64]. In our study, the multi-omic data integration added significant information to boost predictive performance. Strikingly, our findings were observed in blood instead of biopsies containing neuronal cells which would be more tissue-specific for DSPN but are not accessible in large epidemiological cohort studies. Results of this study highlighted the utility of a less invasive blood-based assay to study complex diseases such as DSPN in clinical practice. Although the prediction performance could be improved further by increasing the quantity and quality of data collection and more advanced machine learning development, we believe that using such a model could both be valuable in clinical practice and for the design of future intervention studies. On the one hand, the early identification of people at elevated risk of DSPN could lead to an intensification of (pharmacological and non-pharmacological) risk factor treatment in these people. On the other hand, our model could be used for an enrichment of high-risk individuals in future intervention trials which could reduce required sample size and therefore the costs to assess novel prevention and treatment options. In the long run, our results indicate potentially actionable biomarkers that could be targeted by novel therapy concepts.

One limitation of this study is that our IMML relies substantially on the availability of prior biological knowledge and functional annotation of biomolecules, which thereby reduces the number of evaluated features and may introduce a bias. That being said, incorporating biological knowledge during feature selection increased interpretability, reduced multiple hypothesis testing and utilised cumulatively low-effect size features, overall boosting the model performance. It is our belief that IMML achieves a good balance between predictive power and interpretable DSPN signature, which thereby increases clinical translatability. In addition, validation with external cohorts is currently not feasible due to the uniqueness of the KORA dataset, i.e. deep molecular and longitudinal phenotypic DSPN characterisation, which empowered this study.

One aspect that we were unable to claim are causal relationships due to the inherent limitations of the Kora study design. This is neither addressable in the cross-sectional studies, nor in the prospective segment of our analysis, which concentrates on the occurrence of diabetic sensorimotor polyneuropathy (DSPN), however, the latter sheds light on the temporal associations between risk factors and the onset of DSPN. To delve into the aspect of causality, alternative methodologies are warranted, such as Mendelian randomisation studies conducted in human cohorts or investigations utilising animal models and in vitro studies employing pertinent cell culture models of neurotoxicity. Our findings concerning incident DSPN offer promising candidates for such inquiries subject to further studies.

In summary, we presented the IMML framework which allows studying multifactorial diseases, here exemplified with DSPN. Leveraging IMML, we were able to stratify individuals according to prevalent DSPN status using only clinical variables. More importantly, IMML showed that molecular data is essential to predict the incidence of DSPN, and pathological signatures are detectable in blood samples 6–7 years before disease onset. IMML is capable of integrating sparse multimodal data, and is generalisable to other cohorts and comorbidities. In essence, IMML simplifies the integration and interpretation, thus giving insights in the disease pathophysiology of DSPN, and may navigate the next generation of diagnostic, prevention and treatment strategies of DSPN.

## Data availability

The KORA F4/FF4 cohort dataset is not publicly available due to data protection agreements to protect patient confidentiality. However, data access for research purposes can be requested via the KORA platform at https://helmholtz-muenchen.managed-otrs.com/external/. Data of the results shown in the main figures is in Supplementary Data 3.

## Code availability

The analysis code of this paper is available at https://github.com/phngbh/DSPN [65]. The developed IMML framework is available at https://github.com/phngbh/IMML [66].

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

## Acknowledgements

The KORA study was initiated and financed by the Helmholtz Zentrum München—German Research Centre for Environmental Health, which is funded by the German Federal Ministry of Education and Research and by the State of Bavaria. Furthermore, KORA research was supported within the Munich Centre of Health Sciences (MC-Health), Ludwig-Maximilians-Universität, as part of LMUinnovativ. The German Diabetes Centre is supported by the Ministry of Culture and Science of the State of North Rhine-Westphalia and the German Federal Ministry of Health. This study was supported in part by a grant from the German Federal Ministry of Education and Research to the German Centre for Diabetes Research (DZD). The funders had no role in the study design, data collection and analysis, decision to publish, or preparation of the paper.

## Author contributions

C.H. and M.P.M. conceptualised the project. H.P., B.T., J.A., G.K., M.W., C.G., A.P., K.S., G.J.B., W.R., H.G. and D.Z. acquired and pre-processed the raw data. P.B.H.N. performed exploratory data analysis, developed the machine learning framework and visualised results. P.B.H.N., D.G., H.M., C.H. and M.P.M. derived biological interpretation. P.B.H.N. wrote the paper. M.P.M., C.H. and M.R. revised the paper.

## Funding

## Competing interests

The authors declare no competing interests.

## Additional information

[1]Institute of Computational Biology, Helmholtz Munich, 85764 Neuherberg, Germany. [2]Faculty of Biology, Ludwig-Maximilians University Munich, 82152 Martinsried, Germany. [3]German Center for Diabetes Research (DZD), 85764 Neuherberg, Germany. [4]Institute for Clinical Diabetology, German Diabetes Center, Leibniz Center for Diabetes Research at Heinrich Heine University Düsseldorf, 40225 Düsseldorf, Germany. [5]Institute of Neurogenomics, Helmholtz Munich, 85764 Neuherberg, Germany. [6]Institute of Human Genetics, Technical University Munich, 80333 Munich, Germany. [7]Institute of Epidemiology, Helmholtz Munich, 85764 Neuherberg, Germany. [8]Institute for Medical Information Processing, Biometry and Epidemiology, Ludwig-Maximilians University Munich, 81377 Munich, Germany. [9]Institute of Experimental Genetics, Helmholtz Munich, 85764 Neuherberg, Germany. [10]Department of Biochemistry, Yong Loo Lin School of Medicine, National University of Singapore, Singapore 117597, Singapore. [11]Institute of Biochemistry, Faculty of Medicine, University of Ljubljana, 1000 Ljubljana, Slovenia. [12]Institute of Bioinformatics and Systems Biology, Helmholtz Munich, 85764 Neuherberg, Germany. [13]Research Unit Molecular Epidemiology, Helmholtz Munich, 85764 Neuherberg, Germany. [14]Department of Physiology and Biophysics, Weill Cornell Medicine - Qatar, Education City, Doha 24144, Qatar. [15]Department of Endocrinology and Diabetology, Medical Faculty and University Hospital Düsseldorf, Heinrich Heine University Düsseldorf, 40225 Düsseldorf, Germany. [16]Institute of Biometrics and Epidemiology, German Diabetes Center, Leibniz Center for Diabetes Research at Heinrich Heine University Düsseldorf, 40225 Düsseldorf, Germany. [17]Department of Biochemistry and Pharmacology, Bio21 Molecular Science and Biotechnology Institute, The University of Melbourne, Parkville, Victoria, Australia. [18]These authors contributed equally: Christian Herder, Michael P. Menden. ✉e-mail: Christian.Herder@ddz.de; michael.menden@unimelb.edu.au

