## [Peer Review File · Communications Medicine]

Reviewers' comments:

Reviewer #1 (Remarks to the Author):

The article's title is:

The Interpretable Multimodal Machine Learning (IMML) framework reveals pathological signatures of distal sensorimotor polyneuropathy

This is an analysis of the Kora dataset.

Kora dataset, with 900 controls and almost 200 patients with diabetic neuropathy, is certainly a very valuable asset. First of all a disclaimer: I am not able to assess the methods in this manuscript, and an expert in such techniques should be asked for this. If the article is for subspecialty experts only, then I am the wrong reader. If physicians interested in diabetic neuropathy are supposed to read it, then it needs to be rewritten in a more understandable way.

[EDITOR'S NOTE: Please do ensure the manuscript is worded in such a way that a clinician will be able to understand it given clinicians represent a large part of our target audience.]

The title is difficult to read, and I am not sure what the authors' intention is.

In the abstract, they state:

“we developed the Interpretable Multimodal Machine Learning (IMML) framework for empowering DSPN prevalence and incidence prediction based on sparse multimodal data.”

From this, I understand they want to predict prevalence and incidence.

However, the result section of the abstract reads:

“Important features included up-regulation of proinflammatory cytokines, down-regulation of SUMOylation pathway and essential fatty acids, thus yielding insights in the disease pathology. These may become biomarkers for incident DSPN and guide prevention strategies.”

So, was the aim to identify biomarkers? In any case, the aim should be clearly stated.

I do not understand how “incident DSPN” was defined and ascertained.

One of the main results is that age, waist circumference and height were different in the diabetic neuropathy group. Inflammatory proteins were also more frequent. Both are not surprising, but still would be valuable results, if they can really predict DSPN. The main message should be stated. Which factors predict DSPN? How early does this prediction work? How certain is this prediction? Which of these factors are modifiable? Can the authors clearly distinguish predictive factors from accompanying factors of the disease?

Reviewer #2 (Remarks to the Author):

The authors use multi-modal data including clinical and multi-omic data to predict DSPN diagnosis/prevalence and incidence via elastic net. The modality features were selected by the top-ranked features up to the highest GSEA enrichment score and the modality is selected via 5-fold cross-validation. Interestingly, the molecular data contribute significant improvement to the DSPN incidence prediction but not to the DSPN prevalence. While the overall results are interesting, the method description lacks details.

Major comments:

- elastic net does not account for the different distributions or numerical scales of the multi-modal features. What's the rationale of using it? For example, random forest may be a better and more robust method here.

- There are many Reactome gene sets, and not all of the Reactome gene sets are relevant. What gene set was used for the lead edge analysis followed by feature selection up to the leading edge enrichment peak? What's the ID for the gene set? Is it just one gene set or union of multiple gene sets?

- Line 502: How to convert SNPs to genes? Aggregate SNP effects over each nearby genes?

- Line 513: "Features were considered for further selection process of all models achieving an AUROC > 0.5" I don't understand this. Shouldn't it be the features of non-zero elastic net regression coefficients being selected?

- Line 515: Briefly describe how does RRA aggregate the gene lists. What does the FDR control?

- What are the setting for the alpha and lambda in the elastic net? Did you choose them by 5-fold CV?

- Line 532: What is "model complexity" defined here?

- Line 546: more details needed. What are the individual classifiers on each modality (elastic net?)? What is the ensemble model in the meta-analysis?

Minor comments:

- In Fig 1b, label feature category name on the left of the bars
- In Fig 1, some extra text erroneously showing up on the right side.
- Fig1h font is too small.

Reviewer #3 (Remarks to the Author):

In this study, an interpretable multimodal machine learning (IMML) framework is introduced to systematically integrate multi-omics and clinical data to predict the prevalence and future onset of distal sensorimotor polyneuropathy (DSPN). The results suggest that adding omics modalities to clinical data can improve the prediction of the onset of DSPN about 6 years in advance. Furthermore, the IMML framework allows for modality-specific (and integrated) interpretation and the paper makes a genuine attempt to further our understanding of DSPN.

The paper is clearly written and the methods are well explained. While the paper presents solid empirical work on both the feature engineering and machine learning aspects of the question, the innovative aspects of the work are less obvious. The idea of explicit feature engineering in biomedical data sets to increase interpretability is not necessarily novel. Furthermore, one of the major biological findings related to inflammatory responses is corroborated by previous work done on the exact same data set. Overall, the methodological and biological advances in this paper are modest. In addition, I outline some methodological issues with the work, that further reduce my enthusiasm for its acceptance.

MAJOR:

- The fact that FFS performs comparably to ensemble stacking in Supp. Fig. S2, regardless of the choice of thresholding or GSEA for step 1, greatly undermines the overall IMML approach and this study in general. It appears that one could naively train an ensemble model without GSEA features and achieve similar performance on incident DSPN as that of IMML. Interpretability of the ensemble model could be extracted in a post hoc manner through existing feature importance methods in machine learning such as SHAP.
- It is interesting that molecular features worsen performance over clinical variables for prevalent DSPN and improve performance for incident DSPN. But is this that surprising, given that the model is specifically trained for the incident DSPN prediction task, by optimizing over the FF4 data set? How does a model trained on F4 data do when directly applied on FF4 data as a test set? In my opinion, this is the more appropriate model to evaluate for incident DSPN prediction. In reality, one would have access to features up until a certain timepoint and would need to make predictions in the future from that timepoint. In the paper, an FF4-optimized model assumes that we have access to labels from the future.

- The constraints of not having complete data for a vast majority of samples are understandable but I am concerned that there may be implicit biases in choosing the test set to contain only those individuals for whom all data modalities were available. Without any comparisons of the demographics and metadata associated with the samples for the held-out test set vs. the overall training data, it is unclear if the performance results and gains are generalizable and robust or not. One example of this is that age appears as a significant feature for both prediction tasks. Perhaps the model is learning to distinguish young patients from older ones as a proxy for distinguishing between DSPN and unaffected individuals?

- Although the framework here drastically reduces the feature space used, there still seems to be an issue with the samples sizes being much smaller than the number of features. While elastic net addresses this to an extent, it would be helpful to compare against other machine learning algorithms designed to address this issue, e.g. SVMs.

MINOR:

- In Fig. 1, panels “c” through “h”, it would be good to explicitly state whether the sample numbers come from F4, FF4 or both.

- In figures 2c and 3c, it appears that the both case and control score distributions peak in the 0.3-0.5 range. This suggests that the scores may be miscalibrated and one way to diagnose this would be through calibration plots. Having said this, the framing of two tasks: prevalent and incident DSPN, suggests that this whole study involves positive-unlabeled learning and calibration in such situations is not necessarily straightforward to interpret.

- Although the framework here drastically reduces the feature space used, there still seems to be an issue with the samples sizes being much smaller than the number of features. While elastic net addresses this to an extent, SVMs and random forests may provide some gains predictive performance, if there is reason to believe, that the cases and controls are non-linearly separable in feature space.

- In the Methods section (“Gene set enrichment analysis”), an FDR threshold of 20% seems permissive and rather arbitrary. What is the rationale for this cutoff?

- In the Methods section (“Extraction of leading-edge genes”), there is some inconsistency between the text and Figure 1e. From the figure, only the clinical features are selected out using elastic nets but the text suggests that the molecular features are also subject to the same procedure. Can this be made more clear?

- In the Methods section (“Iterative forward feature selection”), it would be helpful if the nested cross-validation could be explained more clearly. As I understand it, the outer loop is not strictly cross-validation but 100 bootstrap samples of the 80-20 split.

Reviewer #1: Neurologist, neuropathic pain

The article's title is:

The Interpretable Multimodal 1 Machine Learning (IMML) framework reveals pathological signatures of distal sensorimotor polyneuropathy

This is an analysis of the Kora dataset. Kora dataset, with 900 controls and almost 200 patients with diabetic neuropathy, is certainly a very valuable asset. First of all a disclaimer: I am not able to assess the methods in this manuscript, and an expert in such techniques should be asked for this. If the article is for subspecialty experts only, then I am the wrong reader. If physicians interested in diabetic neuropathy are supposed to read it, then it needs to be rewritten in a more understandable way.

Thanks a lot for this invaluable feedback. We aim to 1) deliver a novel interpretable machine learning framework for pathological signatures of DSPN, as well as 2) making it accessible to clinicians and neurologists. In order to address the second point, we have majorly revised our manuscript to increase readability and clarified clinically ambiguous machine learning terminology. Please see our detailed response below.

- **The title is difficult to read, and I am not sure what the authors' intention is.**

Our apologies, there was a typo in the title, which has been corrected from

"The Interpretable Multimodal 1 Machine Learning (IMML) framework reveals pathological signatures of distal sensorimotor polyneuropathy."

to

"The Interpretable Multimodal Machine Learning (IMML) framework reveals pathological signatures of distal sensorimotor polyneuropathy."

The aim of this title is highlighting the novelty of the machine learning framework and its use case in distal sensorimotor polyneuropathy.

- **In the abstract, they state: "we developed the Interpretable Multimodal Machine Learning (IMML) framework for empowering DSPN prevalence and incidence prediction based on sparse multimodal data." From this, I understand they want to predict prevalence and incidence. However, the result section of the abstract reads: "Important features included up-regulation of proinflammatory cytokines, down-regulation of SUMOylation pathway and essential fatty acids, thus yielding insights in the disease pathology. These may become biomarkers for incident DSPN and guide prevention strategies." So, was the aim to identify biomarkers? In any case, the aim should be clearly stated.**

Thanks for pointing this out. We agree that the core-objectives are predicting incidence and prevalence of DSPN. Noteworthy, the IMML framework is interpretable, thus further empowers biomarker detection. Therefore, our model does both. In order to highlight this, we have adjusted the abstract accordingly:

"..., we developed the Interpretable Multimodal Machine Learning (IMML) framework for predicting DSPN prevalence and incidence based on sparse multimodal data. Exploiting IMMLs interpretability further empowered biomarker identification. ..."

Furthermore, we clarified that we investigated the predictive and interpretable feature of the incidence prediction with respect to implications for the pathophysiology of DSPN:

“... Important and interpretable features of the prediction of incident DSPN included up-regulation of proinflammatory cytokines, down-regulation of SUMOylation pathway and essential fatty acids, thus yielding insights in the disease pathophysiology. ...”

- I do not understand how “incident DSPN” was defined and ascertained.

Thanks for this comment, we have expanded the **Methods** section for clarification:

“We used the examination part of the Michigan Neuropathy Screening Instrument (MNSI) score to assess the status of DSPN for all participants of KORA F4 and KORA FF4, as described previously (Herder et al., 2017). In the MNSI assessment, we evaluated the appearance of feet (normal or any abnormalities such as dry skin, calluses, infections, fissures, or other irregularities), foot ulceration, ankle reflexes, and vibration perception threshold at the great toes which was assessed with the Rydel-Seiffer graduated C 64 Hz tuning fork (Feldman et al., 1994). A scoring system ranging from 0 (indicating normal in all aspects) to a maximum of 8 points was used. The normal limits for vibration perception threshold, adjusted for age, were determined based on the method outlined by Martina et al. The MNSI score also included the bilateral examination of touch/pressure sensation using a 10-g monofilament (Neuropen) (Boyraz & Saracoglu, 2010). Therefore, the total MNSI score ranged from 0 (indicating normal in all aspects) to a maximum of 10 points. Considering the advanced age of the participants and the inclusion of the monofilament examination, we defined distal sensorimotor polyneuropathy (DSPN) as a score of equal or higher than 3 points (Herder et al. Diabetes 2018). Thus, participants with an MNSI score ≥ 3 in KORA F4 were considered as prevalent DSPN cases, whereas participants without DSPN in KORA F4 (MNSI < 3) but MNSI ≥ 3 in KORA FF4 were considered as incident cases. This definition meets the minimal diagnostic criteria for possible DSPN, as outlined by the Toronto Diabetic Neuropathy Expert Group (Tefsaye et al., 2010).”

Using the above scoring scheme, we were able to identify people with and without DSPN in F4 and FF4 time points in the KORA cohort. An incident DSPN occurred when a person without DSPN at F4 developed DSPN at FF4. Now, to clarify and highlight the incident DSPN definition, we describe this within a dedicated paragraph:

“Using this criterion, for prevalent DSPN analysis, among 1,091 out of 1,133 individuals having MNSI scoring records, there were 188 cases and 903 controls. For incident DSPN analysis, we only considered the 903 controls in the KORA F4, and examined their progression of DSPN status in the KORA FF4. Among these, we excluded 378 individuals that either did not participate or lacked MNSI scoring records in the KORA FF4. For the incident DSPN analysis, the remaining 521 participants were split into 131 DSPN cases and 394 controls. For both predictions of prevalent and incident DSPN, we only leveraged clinical and molecular features collected at the early time point of KORA F4.”

- One of the main results is that age, waist circumference and height were different in the diabetic neuropathy group. Inflammatory proteins were also more frequent. Both are not surprising, but still would be valuable results, if they can really predict DSPN. The main message should be stated. Which factors predict DSPN?

Thanks for highlighting this. We have revised the **Discussion** to further explain predictive and interpretable features of DSPN prevalence:

“..The analysis of prevalent DSPN (classification of case-control in the F4 population) suggested that the clinical model (using only clinical variables) outperformed the concatenated models (using clinical + molecular variables) in prediction. Then, feature analysis of the clinical model suggested that age, height, neurological illness, and waist circumference were the most important factors that influence the prediction of prevalent DSPN. [...]. In summary, for prevalent DSPN, our analysis is confirmatory of previous studies with respect to these clinical variables. However, here we report the clinical variables in the context of a comprehensive multi-modality analysis of DSPN prevalence, thus adding another layer of information to the model. “

Furthermore, we clarify the predictive features of DSPN incidence:

“... In the case of incident DSPN prediction, the molecular variables added prediction value as they helped improve the prediction performance (higher AUROC values) compared to the clinical model alone. Feature analysis detected multiple important and potentially actionable biomarkers such as inflammatory proteins, SUMOylation-related transcripts and essential fatty acids. Although the association between inflammatory proteins and incident DSPN has been reported before (Herder Diabetes Care 2017, Herder Diabetes 2018), there are as yet no data from population-based studies such as ours implicating SUMOylation-related transcripts and essential fatty acids in the development of DSPN so that these findings are novel and merit further investigation in other cohorts. Additionally, none of these biomarkers and pathways has been reported before in the context of our novel multi-modality analysis of DSPN incidence...”

- **How early does this prediction work?**

We are limited by the design of the KORA F4 and FF4 studies. Participation in both studies was on average 6.5 years (± 0.2 years) apart with no data on incident DSPN in the meantime. To further clarify this point, we expanded the manuscript as the following:

“The KORA cohort contains repeated assessment of DSPN status using identical examination methods at two timepoints, which allows studying both prevalent and incident DSPN. The fact that the mean follow-up time was 6.5 years and that we do not have data on DSPN diagnosis between both studies means that our data cannot be extrapolated to considerably shorter or longer time-periods than 6.5 years. It is possible that different variables may be more powerful for short-term or very long-term prediction of DSPN which needs to be addressed in future studies.”

- **How certain is this prediction?**

In this study we use AUROC (area under the ROC curve) to assess the prediction power of our model. An AUROC of 0.5 or less reflects random or worse than random predictions, respectively. Besides reporting median AUROC values in our manuscript, now we added 95% confidence interval to the **Results** section as well as **Figure 2 & 3** to provide information on the precision of the predictive power of our model:

*“...The clinical model had a median area under the receiver operating curve (AUROC; **Methods**) value of 0.752 with an interquartile range (IQR) of 0.686-0.821 and 95% confidence interval (CI) of 0.733-0.770, whilst the best performing model with molecular data only achieved a median AUROC of 0.583 with IQR of 0.539-0.627. This suggested that clinical variables alone are sufficient to stratify individuals with and without DSPN.”*

“... Identical to previous analysis, first we built a baseline model leveraging clinical features alone, which achieved a median AUROC of 0.603 with an IQR of 0.543-0.676 and 95% CI of 0.588-0.624. This was significantly outperformed by adding either one or two additional molecular data modalities (**Fig. 3a,b; Supp. Fig. S13**), i.e. median AUROC of 0.678 with an IQR of 0.612-0.752 and 95% CI of 0.652-0.692 and AUROC of 0.700 with IQR of 0.651-0.774 and 95% CI of 0.686-0.722, respectively (Wilcoxon rank sum test, $p = 1.9e-16$ and $p = 2.9e-11$, respectively). We observed that inflammatory proteins were >80% the first picked molecular layer, followed by metabolites, whilst SNPs and CpG sites seemed to carry the least predictive information (**Fig. 3a**). The performance tended to saturate at 3-modality complexity as adding more modalities did not significantly improve the performance anymore (Wilcoxon rank sum test, $p=0.95$), i.e. 4-modality models had a median AUROC of 0.714 with an IQR of 0.640-0.774 and 95% CI of 0.684-0.720 (**Fig. 3b**)...”

- **Which of these factors are modifiable?**

With respect to prevalent DSPN, we identified mostly clinical variables of which waist circumference is modifiable, as shown in the **Results** and **Discussion**:

“The analysis of prevalent DSPN (classification of case-control in the F4 population) suggested that the clinical model (using only clinical variables) outperformed the concatenated models (using clinical + molecular variables) in prediction. Then, feature analysis of the clinical model suggested that age, height, neurological illness, and waist circumference were the most important factors that influence the prediction of prevalent DSPN. Age and height have been reported to be associated with prevalent DSPN (Ziegler et al., 2022). The neurological comorbidity status of patients is not used to classify DSPN yet, however, there might be an intrinsic neurological mechanism that links DSPN to other neurological illnesses. Finally, waist circumference is strongly correlated with BMI, which has been reported to be a risk factor for developing DSPN (Fakkel et al., 2020). From a clinical perspective it is worth mentioning that only waist circumference represents a modifiable risk factor which emphasizes the role of obesity prevention and treatment also in the context of DSPN.”

Furthermore, we expanded our **Discussion** on modifiable factors that are important for incident DSPN prediction. First we discuss inflammatory effects:

“Remarkably, inflammatory effects were observed in the blood samples prior to disease onset. Thus, the predictive pro-inflammatory cytokines, chemokines and transmembrane proteins observed in this study could represent modifiable risk factors and therefore therapeutic targets for disease prevention. For example, salicylate was reported in many studies to have inhibitory effects on production of cytokines and chemokine. In addition, novel treatment approaches targeting IL-1 beta-related mechanisms have been demonstrated to reduce subclinical inflammation and have beneficial effects on cardiometabolic risk, and may be generalisable for DSPN. Beyond pharmacological approaches to attenuate subclinical inflammation, it is important to emphasise that subclinical inflammation is triggered by a range of other modifiable risk factors such as high-calorie diet, certain nutrients, physical inactivity, obesity, psychosocial stress and sleep disturbances so that lifestyle changes represent another option for intervention (Furman D et al., 2019).”

Furthermore, we propose the actionable follow-ups for the SUMO pathway:

“The transcriptomic layer also gained attention as one of the most important predictors of DSPN. Particularly, significant down-regulation of the small ubiquitin-related modifier (SUMO) pathway was consistent with a recent study, which demonstrated SUMO posttranslational modifications are involved in glycolysis. Furthermore, the tricarboxylic acid (TCA) cycle plays a crucial role in maintaining important metabolic processes in sensory neurons, and deficiency of SUMO activity causes damaging effects which may specifically contribute to DSPN pathogenesis. Although this enrichment analysis has to be interpreted with caution due to the small sample size, it is noteworthy that oxidative stress and inflammation have been proposed as mediators linking hyperglycaemia and impaired SUMOylation in diabetic polyneuropathy and that aberrant SUMOylation has also been implicated in the aetiology of neurodegenerative diseases. Thus, this finding extends the aforementioned results on inflammation, corroborates other studies and may point towards another mechanism how DSPN risk could be targeted by addressing modifiable risk factors leading to inflammation and oxidative stress.”

And finally, we discuss the actionability of fatty acids:

“ [...] Overall, caprate, linolenate and adrenate have not been linked to DSPN in detailed investigations but nevertheless highlight the possibility that they should be modifiable risk factors that could be modulated by specific dietary interventions or dietary supplements. Importantly, experimental results suggested that PUFA might be a potential agent to treat DSPN (Tao et al., 2008; Duran et al., 2022), subject to future studies focussing on high-risk individuals assessing the potential preventive and therapeutic properties of dietary fatty acids in this context. ”

- **Can the authors clearly distinguish predictive factors from accompanying factors of the disease?**

We thank the reviewer for this important question. With cross-sectional studies, this issue cannot be addressed. The prospective part of our analysis focussing on incident DSPN can clarify the aspect of temporal relationships between risk factors and onset of DSPN but does not allow for causal inference. The aspect of causality would need approaches such as Mendelian randomisation studies in human cohorts or studies based on animal models or *in vitro* studies using relevant cell culture models of neurotoxicity. Our findings on incident DSPN provide good candidates for such analyses, but we believe that adding them would be beyond the scope of our manuscript.

“One aspect that we were unable to claim are causal relationships due to the inherent limitations of the KORA study design. This is neither addressable in the cross-sectional studies, nor in the prospective segment of our analysis, which concentrates on the occurrence of Diabetic Sensorimotor Polyneuropathy (DSPN), however, the latter sheds light on the temporal associations between risk factors and the onset of DSPN. To delve into the aspect of causality, alternative methodologies are warranted, such as Mendelian randomization studies conducted in human cohorts or investigations utilizing animal models and in vitro studies employing pertinent cell culture models of neurotoxicity. Our findings concerning incident DSPN offer promising candidates for such inquiries subject to further studies.”

Reviewer #2: Computational, deep learning, multimodal data, complex traits

- **While the overall results are interesting, the method description lacks details**

Thanks for acknowledging the novelty of our finding. We agree that extending the **Methods** section is beneficial for reproducibility, thus have comprehensively extended it. Please see the answers to your comments below, as well as the revised **Methods** section for more details.

- **elastic net does not account for the different distributions or numerical scales of the multi-modal features. What's the rationale of using it? For example, random forest may be a better and more robust method here.**

Elastic net was employed due to its inherent interpretability. Particularly, feature importance could be easily extracted by computing t-statistics of the models' beta values.

We agreed that the multi-modal features are heterogenous and come in different numerical scales. To clarify this, we have adjusted the text as the following.

"[...] Each molecular layer was standardized before downstream analysis by computing the z-score, which accounts for different distributions and numerical scales of features. Our analysis pipeline pre-processed the data in a modality-specific manner, as shown below."

Furthermore, we agree that using nonlinear algorithms may be beneficial, thus, included them in a benchmarking and compared performance of predictions:

*"To select the machine learning algorithm for DSPN prediction, we compared the predictive performance of elastic net, random forest and support vector machine, the latter leveraged linear and radial kernels (**Supp. Fig. S9**). For this we performed 100 matched resamples with forward feature selection. Elastic net outperformed the other three machine learning algorithms in both prevalent DSPN (**Supp. Fig. S9a-d**) and incident DSPN predictions (**Supp. Fig. S9e-h**). Best performances in prevalent DSPN (AUROCs of 0.737) and incident DSPN (AUROCs of 0.708) predictions were observed at 1-modality and 3-modality complexity, respectively. Notably, none of the other machine learning algorithms reached AUROC higher than 0.700 at any model complexity."*

Supplementary Figure S9: Prediction performance of four different machine learning algorithms. Here we compare the predictive power of (a-d) prevalent DSPN and (e-h) incident DSPN. We benchmarked (a,e) elastic net (glmnet), (b,f) random forest (rf), and support vector machine with (c,g) radial (svmRadial) and (d,h) linear kernel (svmLinear).

- There are many Reactome gene sets, and not all of the Reactome gene sets are relevant. What gene set was used for the lead edge analysis followed by feature selection up to the leading edge enrichment peak? What's the ID for the gene set? Is it just one gene set or union of multiple gene sets?

We have expanded our description accordingly. During the feature selection, we implemented gene set enrichment analysis using Reactome gene sets as reference biological information. We did this in a non-bias and data driven manner so all the gene sets in the database were considered:

“...In all cases, we included the full set of Reactome signaling pathways at the lowest levels of pathway hierarchy to avoid redundancy, and at the time, ensure full unbiased coverage (Supp. Table S5)...”

For further increasing transparency, we created a supplemental table containing all pathway definitions (**Supp. Table S5**), as well as downstream analysis with Robust Rank Aggregation algorithm to extract the most significant gene sets (**Supp. Table S3 & S4**).

- **Line 502: How to convert SNPs to genes? Aggregate SNP effects over each nearby genes?**

Thanks for this question. We mapped SNPs according to their proximity to genes, however, we further refined the method for gene set analysis for genomic data (and also DNA methylation data) as the following:

“... For genomic data, we leveraged the MAGMA software (de Leeuw CA et al.) to estimate the gene effect and subsequently perform gene set analysis. First, we annotated SNPs according to nearby genes (2 kb upstream and 0.5 kb downstream), and consecutively used MAGMA to estimate the gene effect on the phenotype, taking into account the SNPs that were mapped to this gene. MAGMA estimated the gene effect by first conducting principal component analysis (PCA) using all SNPs linked to this gene, and afterwards used PCs to train a linear regression model predicting the phenotype. Finally, MAGMA computed the gene’s p-value with F-test, and converted these to Z-values for the gene set analysis leveraging a linear regression model.

For methylomic data, we used the methylRRA method (Ren X et al.) to perform gene set enrichment analysis (GSEA) on the CpG probes. First, this required a differential expression analysis on the probes using the R package limma, followed by using the ranked list of p-values as input for methylRRA. To this end, methylRRA computed a p-value for each gene leveraging the ranking of all CpGs annotated to that gene by implementing Robust Rank Aggregation algorithm. Consequently, the p-values were transformed into z-scores and were used for the GSEA to extract significant gene sets.”

- **Line 513: “Features were considered for further selection process of all models achieving an AUROC > 0.5” I don’t understand this. Shouldn’t it be the features of non-zero elastic net regression coefficients being selected?**

Thanks for pointing this out. In order to clarify, we adjusted the manuscript as below:

“Leading-edge genes in upregulated gene sets are all genes from the beginning of the ranked gene list until the enrichment score (ES). In contrast, in case of down-regulated gene sets, leading-edge genes are from the ES to the end of the ranked gene list. Here, we leveraged 80% of all data for each of the 100 stratified resamples, did GSEA, extracted the leading edge molecules to train an elastic net model, and finally tested the model prediction on the left out remaining 20% of samples. For aggregating results of these 100 stratified resamples, we only considered predictive models (AUROC > 0.5), and leveraged a Robust Rank Aggregation (RRA) algorithm with a false discovery rate (FDR) cutoff of 5%, which delivered a union of leading-edge gene sets. Afterwards, a GSEA was conducted on the union of leading-edge gene sets to extract the final consensus significant gene sets and leading edge molecules, which were subject to final model training.”

- **Line 515: Briefly describe how does RRA aggregate the gene lists. What does the FDR control?**

Thanks for this comment. We added a section in the **Methods** on robust rank aggregation:

“Robust Rank Aggregation

We leveraged the implementation of Robust Rank Aggregation of Kolde et al. [26]. The molecules/molecule sets were ranked according to p-values, leading to a different ranked list per cross-validation/resampling run. Then, the rank distribution of each molecule/molecule set across all lists was tested against the random ranking distribution generated by permutation with the null hypothesis that there was no difference between the two distributions. The p-values of the test were adjusted for multiple hypothesis testing by multiplying the number of tested lists and additionally adjusted for the number of tested molecules/molecule sets by Benjamini-Hochberg method.”

- **What are the setting for the alpha and lambda in the elastic net? Did you choose them by 5-fold CV?**

Thanks, we revised the manuscript to add more details regarding this:

“The iterative forward feature selection (FFS) integrates multiple data modalities. It is based on a nested cross-validation leveraging elastic net models with weighted log-loss function to overcome class imbalances [...]. We tuned elastic net’s hyperparameters alpha and lambda by grid search of 20 alphas and lambdas in range [0,1], resulting in 400 parameter sets. The chosen hyperparameter combination was the one having best mean performance across 5-fold cross-validations”

- **Line 532: What is “model complexity “defined here?**

Thanks for pointing this out. The model complexity here means the number of modalities as input to the model. The more modalities, the higher complexity:

“For the second step, i.e. final model training and multimodal data integration, we leveraged the short-listed features from the analyses above. The final model was trained with an embedded feature selection whilst balancing model complexity. In the context of this study, we refer to the number of input modalities as model complexity ...”

- **Line 546: more details needed. What are the individual classifiers on each modality (elastic net?)? What is the ensemble model in the meta-analysis?**

Thanks for the comment, we have updated the **Methods** section accordingly:

“Benchmarking of feature selection and integration methods

For feature selection, we compared GSEA with the conventional thresholding methods. For feature integration, we benchmarked FFS, data concatenation and ensemble stacking approaches. Thus, in total there were six combinations of methods to compare. For the thresholding method, we implemented differential expression analysis using limma and selected features having p-values<0.05.

Regarding feature integration, we benchmarked the FFS with ensemble stacking and feature concatenation. The latter concatenated all features into a single matrix before training the model. The ensemble stacking approach leveraged 100 independent runs with stratified resampling. This is, we generated 100 sets of stratified resamples, each

consisting of 80% training and 20% test set (i.e. outer loop). Within each 80% training set, we further divided the data into 5-fold cross validation sets (i.e. inner loop). For each iteration of the inner loop, we trained an elastic net model on four out of five validation sets and made predictions on the remaining validation set. After five iterations, we obtained the predictions of all samples for that inner loop (corresponding to the 80% training set from the outer loop). We used this together with the ground truth (80% outer train) to train an elastic net meta model in the outer loop, and consecutively tested the predictive performance on the remaining 20% test set. Importantly, the test sets were never used for any parameter optimisation nor training, and only leveraged for unbiased performance evaluation. This process was repeated for each data modality. For the feature analysis we implemented Robust Rank Aggregation on the meta models of the ensemble stacking across 100 resamplings, then extracted the individual feature importance. ”

We also added the following pseudocode for further explaining the ensemble stacking process.

```

1: for run = 1, 2, ... 100 do
2:   (Trainrun, ytrain) ← Resample 80% data, (Testrun, ytest) ← remaining
   20%
3:   # Train single-modality model in each run
4:   Initialize modality-specific prediction collection matrix Predictionsrun
5:   for m = 1, 2, ..., 7 do
6:     train a modality-specific model modelrunm with Trainrunm
7:     predictionrunm = modelrunm.predict(Testrunm)
8:     Predictionsrun[m].append(predictionrunm)
9:   end for
10:
11:
12:   # Train the Meta-model
13:   Initialize modality-specific prediction collection matrix Predictionscv
14:   for cv = 1, 2, ..., 5 do
15:     traincv ← 80% of Trainrun
16:     valcv ← 20% of Trainrun
17:     for m = 1, 2, ..., 7 do
18:       train a modality-specific model modelm with traincvm
19:       predictioncvm = modelm.predict(valcvm)
20:       Predictionscv[m].append(predictioncvm)
21:     end for
22:   end for
23:   PredictionscvT is in shape [#Trainrun, 7]
24:   PredictionsrunT is in shape [#Testrun, 7]
25:   Train the Meta-model with PredictionscvT with ytrain
26:   Test the Meta-model on PredictionsrunT with ytest
27: end for

```

Reviewer #3: Computational, machine learning, genomics

- In this study, an interpretable multimodal machine learning (IMML) framework is introduced to systematically integrate multi-omics and clinical data to predict the prevalence and future onset of distal sensorimotor polyneuropathy (DSPN). The results suggest that adding omics modalities to clinical data can improve the prediction of the onset of DSPN about 6 years in advance. Furthermore, the IMML framework allows for modality-specific (and integrated) interpretation and the paper makes a genuine attempt to further our understanding of DSPN. The paper is clearly written and the methods are well explained.

We appreciate this kind evaluation.

- **While the paper presents solid empirical work on both the feature engineering and machine learning aspects of the question, the innovative aspects of the work are less obvious. The idea of explicit feature engineering in biomedical data sets to increase interpretability is not necessarily novel. Furthermore, one of the major biological findings related to inflammatory responses is corroborated by previous work done on the exact same data set. Overall, the methodological and biological advances in this paper are modest.**

Thanks for this comment. To address this, we would like to further emphasize the novelty of our work.

Firstly, highlighting the robustness of our methodology, we confirmed established and anticipated associations of cross-sectional DSPN predictions. Secondly, to the best of our knowledge, this is the first attempt to leverage multimodal signaling pathway information in an end-to-end prediction task to study incidence DSPN. Furthermore, we systematically leveraged our models to detect individual biomarkers of these phenotypes. Previous DSPN studies, either on the same data set or others, focused on univariate statistical analyses of cross-sectional DSPN, whilst largely neglecting multi-omics integration:

"...Furthermore, we presented an innovative machine learning framework to model incidence of DSPN by integrating multi-omic and clinical data. Previous efforts either focused on classifying the disease in a cross-sectional context, lacked multi-omics integration strategies or exhibited limitations of univariate statistical analyses (Haque et al., Shin et al, Kazemi et al., Jian et al., Dagliati et al. and Herder et al.). In our study, the multi-omic data integration added significant information to boost predictive performance."

Notably, it is the unique study design of the KORA cohort which empowered the DSPN incidence prediction. Now, we further highlight this in our manuscript:

"One strength of our study's design is the utilisation of population-based prospective data from a large cohort (KORA F4). The KORA cohort contains repeated assessment of DSPN status using identical examination methods at two timepoints, which allows studying both prevalent and incident DSPN..."

Finally, we further highlight the novelty of detected biomarkers in the revised manuscript:

"...Feature analysis detected multiple important and potentially actionable biomarkers such as inflammatory proteins, SUMOylation-related transcripts and essential fatty acids. Although the association between inflammatory proteins and incident DSPN has been reported before, there are as yet no data from population-based studies such as ours implicating SUMOylation-related transcripts and essential fatty acids in the development of DSPN so that these findings are novel and merit further investigation in other cohorts. Additionally, none of these biomarkers and pathways has been reported before in the context of our novel multi-modality analysis of DSPN incidence."

- **In addition, I outline some methodological issues with the work, that further reduce my enthusiasm for its acceptance.**

Thank you for your comments. We have addressed them carefully and also adjusted the manuscript accordingly, please see below for details.

- The fact that FFS performs comparably to ensemble stacking in Supp. Fig. S2, regardless of the choice of thresholding or GSEA for step 1, greatly undermines the overall IMML approach and this study in general. It appears that one could naively train an ensemble model without GSEA features and achieve similar performance on incident DSPN as that of IMML. Interpretability of the ensemble model could be extracted in a post hoc manner through existing feature importance methods in machine learning such as SHAP.

In this benchmarking, one of the main messages was that the GSEA-based feature selection performed better than thresholding-based approach, regardless of the choice of multi-modal feature integration methods, which is depicted in now **Supp. Fig. S3**. We revised the manuscript to further explain the algorithms and their selection:

“...We benchmarked three feature integration methods, i.e. forward feature selection (FFS), ensemble and concatenation of all features together (Supp. Fig. S3a; Methods), and observed best performance with GSEA-FFS followed by GSEA-ensemble stacking (Supp. Fig. S3b). When comparing the performance of the FFS and ensemble stacking methods using all modalities and with GSEA as the feature selection approach, the FFS algorithm achieved marginally higher predictive performance (Supp. Fig. S8a). Both methods retained inflammatory proteins as the most predictive features, however, the GSEA-FFS was further able to detect clinically relevant signals from other modalities (Supp. Fig. S8b). Therefore, we implemented an iterative FFS algorithm with resampled cross-validation (Methods).”

We also expanded the Method section to further explain this:

“Regarding feature integration, we benchmarked the FFS with ensemble stacking and feature concatenation. We performed an ensemble stacking approach in 100 independent runs with stratified resampling. This is, we generated 100 sets of stratified resamples, each consisting of 80% training and 20% test set (i.e. outer loop). Within each 80% training set, we further divided the data into 5-fold cross validation sets (i.e. inner loop). For each iteration of the inner loop, we trained an elastic net model on four out of five validation sets and made predictions on the remaining validation set. After five iterations, we obtained the predictions of all samples for that inner loop (corresponding to the 80% training set from the outer loop). We used this together with the ground truth (80% outer train) to train an elastic net meta model in the outer loop, and consecutively tested the predictive performance on the remaining 20% test set. Importantly, the test sets were never used for any parameter optimisation nor training, and only leveraged for unbiased performance evaluation. This process was repeated for each data modality. For the feature analysis we implemented Robust Rank Aggregation on the meta models of the ensemble stacking across 100 resamplings, then extracted the individual feature importance.”

Below is the newly added **Supplemental Figure S8**:

Supplementary Figure S8. Performance of forward feature selection (FFS) and ensemble stacking feature integration methods across 100 stratified resamplings. (a) AUROC of the testing prediction of the two algorithms. P-value of Wilcoxon rank sum test is shown. (b) Important features selected by the GSEA-ensemble stacking (GSEA-Es) and GSEA-FFS methods and their overlapping.

- It is interesting that molecular features worsen performance over clinical variables for prevalent DSPN and improve performance for incident DSPN. But is this that surprising, given that the model is specifically trained for the incident DSPN prediction task, by optimizing over the FF4 data set? How does a model trained on F4 data do when directly applied on FF4 data as a test set? In my opinion, this is the more appropriate model to evaluate for incident DSPN prediction. In reality, one would have access to features up until a certain timepoint and would need to make predictions in the future from that timepoint. In the paper, an FF4-optimized model assumes that we have access to labels from the future.

Thanks for the comment, we suspect this is a misunderstanding. Both prevalent DSPN and incident DSPN predictions exclusively leveraged features of the early F4 timepoint as model input. For the incident DSPN prediction, we additionally leveraged the diagnosis labels (i.e. outcomes) of the later FF4 timepoint. Now, we further clarify this in the manuscript:

“Using this criterion, for prevalent DSPN analysis, among 1,091 out of 1,133 individuals having MNSI scoring records, there were 188 cases and 903 controls. For incident DSPN analysis, we only considered the 903 controls in the KORA F4, and examined their progression of DSPN status in the KORA FF4. Among these, we excluded 378 individuals that either did not participate or lacked MNSI scoring records in the KORA FF4. For the incident DSPN analysis, the remaining 521 participants were split into 131 DSPN cases and 394 controls. Both prevalent and incident DSPN predictions leveraged clinical and molecular features as input collected at the early F4 time point of KORA.”

- The constraints of not having complete data for a vast majority of samples are understandable but I am concerned that there may be implicit biases in choosing the test set to contain only those individuals for whom all data modalities were available. Without any comparisons of the demographics and metadata associated with the samples for the held-out test set vs. the overall training data, it is unclear if the performance results and gains are generalizable and robust or not. One example of this is that age appears as a significant feature for both prediction tasks. Perhaps the model is learning to distinguish young patients from older ones as a proxy for distinguishing between DSPN and unaffected individuals?

If we understand your point correctly, you are concerned about potential biases in selected samples for the feature selection, model training and testing datasets. To this end, we performed PCA using all clinical features and visualized the two datasets in each modality to see if there was obvious segregation. In all modalities, we observed no effect in selection for feature selection and model training sets (**Supp. Fig. S2**), suggesting there was no bias in data partitioning.

Supplementary Figure S2. PCA of clinical features for feature selection and model training datasets. Grey contour plots highlight the model training sets, whilst other colors indicate the feature selection set of the different modalities: (a) Genomics, (b) Transcriptomics, (c) Proteomics, (d) Metabolomics, (e) Methylomics and (f) Clinical data.

We further revised the data partitioning description in the **Results** part of the manuscript:

“...Both subsets of data for feature selection and model training and testing were subject to PCA analysis using clinical information to ensure there was no potential bias in sample selection (Supp. Fig. S2).”

- **Although the framework here drastically reduces the feature space used, there still seems to be an issue with the sample sizes being much smaller than the number of features. While elastic net addresses this to an extent, it would be helpful to compare against other machine learning algorithms designed to address this issue, e.g. SVMs.**

Thanks for this suggestion. We included a benchmarking of elastic net, random forest, SVM with linear and radial kernels:

“To select the machine learning algorithm for DSPN prediction, we compared the predictive performance of elastic net, random forest and support vector machine, the latter leveraged linear and radial kernels (Supp. Fig. S9). For this we performed 100 matched resamples with forward feature selection. Elastic net outperformed the other three machine learning algorithms in both prevalent DSPN (Supp. Fig. S9a-d) and incident DSPN predictions (Supp. Fig. S9e-h). Best performances in prevalent DSPN (AUROCs of 0.737) and incident DSPN (AUROCs of 0.708) predictions were observed at 1-modality and 3-modality complexity, respectively. Notably, none of the other machine learning algorithms reached AUROC higher than 0.700 at any model complexity.”

The performance comparison is shown in **Supp. Fig. S9**:

Supplementary Figure S9: Prediction performance of four different machine learning algorithms. Here we compare the predictive power of (a-d) prevalent DSPN and (e-h) incident DSPN. We benchmarked (a,e) elastic net (glmnet), (b,f) random forest (rf), and support vector machine with (c,g) radial (svmRadial) and (d,h) linear kernel (svmLinear).

- In Fig. 1, panels “c” through “h”, it would be good to explicitly state whether the sample numbers come from F4, FF4 or both.

Thanks for pointing this out. All the samples here were only from the F4 timepoint. Some of the samples had DSPN information in FF4, which was the basis to determine incident DSPN label. In Fig. 1, sample and feature sizes were depicted for incident DSPN analysis. The same information for prevalent DSPN was plotted in Supp. Fig. S1 in the manuscript. In order to clarify, we edited the legend of Fig. 1:

“Figure 1: Workflow of interpretable multimodal framework for feature prioritisation, DSPN classification and disease incidence prediction. (a) Distribution of samples across time points (KORA F4 and FF4), disease status (case or control) at baseline (KORA F4) and follow-up (KORA FF4) and prediction tasks. (b) Number of features stratified according to data modalities. In grey are removed features after pre-processing. (c) Number of samples characterised within each data modality and their overlaps in KORA F4. (d) Fully characterised samples in KORA F4 were exclusively leveraged for (g) the second and final training step, whilst the remaining sparse samples were used for (e) prior feature prioritisation: All molecular features were shortlisted based on differential expression analysis (DEA), gene set enrichment analysis (GSEA) and their leading-edge genes (Methods), whilst clinical features were ranked according to feature importance of elastic

net models. (f) Features for the final training step were selected based on rank aggregation (Methods). (g) The final training set contained 54 DSPN cases and 188 controls in KORA F4. In the second step, elastic net models determined the optimal complexity, features and combination of modalities. These models implemented forward feature selection in a nested cross-validation, using weighted log loss to account for class imbalance, and finally 100 stratified resampling during training and rank aggregation (Methods), thus returning (h) the refined and final model further subject to functional analysis for gaining insights in DSPN pathophysiology.”

- In figures 2c and 3c, it appears that the both case and control score distributions peak in the 0.3-0.5 range. This suggests that the scores may be miscalibrated and one way to diagnose this would be through calibration plots. Having said this, the framing of two tasks: prevalent and incident DSPN, suggests that this whole study involves positive-unlabeled learning and calibration in such situations is not necessarily straightforward to interpret.

Thanks for pointing this out. We solved this by calibrating the predicted probabilities. Particularly, we fit a logistic regression model to the predicted probabilities, using the true binary outcomes as the target variable, taking into account the class imbalance. We added this detail to **Methods** section and add a supplementary figure:

“...The prediction performance of the model was tested by predicting on the outer test set (20% samples). The prediction probabilities were calibrated using the Platt scaling method. In each step, the model adds the next best data modality based on increased performance until all data modalities are included.”

To check if the prediction was calibrated properly, we made calibration curves for both prevalent (Supp. Fig. S10a) and incident DSPN (Supp. Fig. S10b) prediction. The plots showed that the predicted probabilities are mostly close to the true outcomes, except for a few outliers.

Supplementary Figure S10: Calibration analysis. Calibration plots of predicted probabilities for (a) prevalent DSPN and (b) incident DSPN.

Finally we adjusted Fig. 2c & 3c in the manuscript to reflect the calibrated probabilities.

- In the Methods section (“Gene set enrichment analysis”), an FDR threshold of 20% seems permissive and rather arbitrary. What is the rationale for this cutoff?

This lenient cut-off was motivated by allowing the selection of features with lower effect size, which may cumulatively add predictive power when integrated. During modality integration, we implemented an embedded feature selection with elastic net selecting the most informative features. We added the motivation for the lenient threshold to our method description:

“...Finally, the p-values were adjusted for multiple hypothesis testing with false discovery rate (FDR) < 20%, which is a lenient threshold allowing the selection of features with lower effect size, which may add predictive value in multivariate models in later integration steps.”

- **In the Methods section (“Extraction of leading-edge genes”), there is some inconsistency between the text and Figure 1e. From the figure, only the clinical features are selected out using elastic nets but the text suggests that the molecular features are also subject to the same procedure. Can this be made more clear?**

Thank you for pointing this out. We have adjusted the text accordingly:

“Leading-edge genes in upregulated gene sets are all genes from the beginning of the ranked gene list until the enrichment score (ES). In contrast, in case of down-regulated gene sets, leading-edge genes are from the ES to the end of the ranked gene list. Here, we leveraged 80% of all data for each of the 100 stratified resamples, did GSEA, extracted the leading edge molecules to train an elastic net model, and finally tested the model prediction on the left out remaining 20% of samples. For aggregating results of these 100 stratified resamples, we only considered predictive models (AUROC > 0.5), and leveraged a Robust Rank Aggregation (RRA) algorithm with a false discovery rate (FDR) cutoff of 5%, which delivered a union of leading-edge gene sets. Afterwards, a GSEA was conducted on the union of leading-edge gene sets to extract the final consensus significant gene sets and leading edge molecules, which were subject to final model training.”

- **In the Methods section (“Iterative forward feature selection”), it would be helpful if the nested cross-validation could be explained more clearly. As I understand it, the outer loop is not strictly cross-validation but 100 bootstrap samples of the 80-20 split.**

Yes, you are correct. The outer loop was stratified resamples of 80-20 split. We have revised the **Method** section accordingly:

“The iterative forward feature selection (FFS) integrates multiple data modalities. It is based on 100 independent runs of five-fold cross-validation. We tuned elastic net’s hyperparameters alpha and lambda by grid search of 20 alphas and lambdas in range [0,1], resulting in 400 parameter sets. The chosen hyperparameter combination was the one having best mean performance across 5-fold cross-validations. In each run, we randomly sampled 80% of the dataset to perform five-fold cross-validation and the performance was tested with the remaining 20% data. For each fold of model training, elastic net models with weighted log-loss function to overcome class imbalances were implemented. Within the inner loop, a 5-fold cross-validation selected the best data modality to add next. The prediction performance of the model was tested by predicting on the outer test set (20% samples). The prediction probabilities were calibrated using the Platt scaling method. In each step, the model adds the next best data modality based on increased performance until all data modalities are included. “

Reviewers' comments:

Reviewer #1 (Remarks to the Author):

The manuscript has much improved and is better accessible now to the reader.

Reviewer #2 (Remarks to the Author):

Thank the authors for addressing my comments. One more comment is that 'Model complexity' implies the number of parameters or the sophistication of the machine learning model rather than the number of input modalities. I do not recommend use that term here.

Reviewer #3 (Remarks to the Author):

Most of my comments have been addressed satisfactorily and I appreciate the efforts put into strengthening the manuscript. However, one key point still remains. Regarding the distinction between predicting prevalent and incident DSPN, as I understand it, two different models were trained and evaluated in the two-step framework presented here. Even though only features from F4 were used for both models, they were still optimized on two different training data sets. My original question was: what would happen if I trained a single model only on F4 features and data, froze it and used it to make predictions on the FF4 data set? Ideally, the prevalent DSPN model should have learned to weight features such that future DSPN cases within the the F4 controls would be scored higher. In my opinion, this should serve as the baseline to determine if molecular features are truly drivers of incident DSPN prediction. It is also possible the worse-performing model for prevalent DSPN (clinical + molecular features) may end up doing better on incident DSPN prediction, which is a stronger demonstration of predictive success. Essentially, any claims about biological or clinical utility of prediction models should account for situations in which such models would be deployed. Predicting DSPN cases 6.3-6.7 years into the future implies that one does not even have access to even the DSPN labels from the future. If this concern can be addressed or the claim of future prediction toned down, I am comfortable with recommending the publication of this work.

Reviewer #2:

- **Thank the authors for addressing my comments. One more comment is that 'Model complexity' implies the number of parameters or the sophistication of the machine learning model rather than the number of input modalities. I do not recommend use that term here.**

Thank you for your comment. We adjusted the manuscript and systematically changed the terminology to “number of modalities” accordingly. Below are a few examples of textual changes, among others (consistently highlighted in revised manuscript):

*“...We hypothesise that well performing models at **the minimum number of data modalities** will give insights into the disease aetiology of DSPN and its incidence, thus may improve diagnosis and pave the way for prevention strategies...”*

*“...The final model was trained with an embedded feature selection whilst balancing **the number of modalities**...”*

*“...Best performances in prevalent DSPN (AUROCs of 0.737) and incident DSPN (AUROCs of 0.708) predictions were observed at 1-modality and 3-modality **models**, respectively...”*

Reviewer #3:

- **Most of my comments have been addressed satisfactorily and I appreciate the efforts put into strengthening the manuscript. However, one key point still remains. Regarding the distinction between predicting prevalent and incident DSPN, as I understand it, two different models were trained and evaluated in the two-step framework presented here. Even though only features from F4 were used for both models, they were still optimized on two different training data sets.**

Thank you for the positive feedback. Regarding the distinction between prevalent and incident DSPN predictions, you are correct, they are two separate tasks, i.e. the two-step framework was run separately for the two independent models. Both models were trained on the same feature sets from time point F4, however, they used two different output labels.

To clarify that both models use the same features / input from F4, we added further clarification in **Fig. 1a**:

*“(a) Distribution of samples across time points (KORA F4 and FF4), disease status (case or control) at baseline (KORA F4) and follow-up (KORA FF4) and prediction tasks. **Both models were trained on the same set of F4 features but different labels and a subset of samples.**”*

We also further clarified the definition incident and prevalent label:

“...The earlier F4 time point surveyed 1,091 individuals of whom 622 were followed up at the later FF4 time point. We used the established Michigan Neuropathy Screening Instrument (MNSI) to assess and define the DSPN status as described in previous studies [ref 24,25]. Using MNSI, we identified 188 DSPN cases and 903 controls at F4, and 131 controls who developed DSPN between F4 and FF4 (Supp. Table S1, S2). The first machine learning task was to predict DSPN prevalence at F4 (Fig. 1a). The second task was to predict whether controls at F4 will develop incident DSPN during the period from F4 to FF4 (Fig. 1a).”

- **My original question was: what would happen if I trained a single model only on F4 features and data, froze it and used it to make predictions on the FF4 data set? Ideally, the prevalent DSPN model should have learned to weight features such that future DSPN cases within the the F4 controls would be scored higher. In my opinion, this should serve as the baseline to determine if molecular features are truly drivers of incident DSPN prediction. It is also possible the worse-performing model for prevalent DSPN (clinical + molecular features) may end up doing better on incident DSPN prediction, which is a stronger demonstration of predictive success. Essentially, any claims about biological or clinical utility of prediction models should account for situations in which such models would be deployed.**

Thank you for further clarifying your comment. Accordingly, we implemented this baseline model and added it to the manuscript:

“Molecular data improves DSPN incidence prediction

DSPN incidence prediction was strongly enhanced by integrating clinical and molecular data. In contrast to clinical baseline models (Supplemental Figure S13a,b), we observed a strong benefit in leveraging molecular modalities for predicting whether participants of the KORA F4 cohort will develop DSPN or not within the next 6.5 ± 0.2 years (Fig. 3a,b; Supplemental Figure S13c). The baseline DSPN incidence model using clinical features alone achieved a median AUROC of 0.603 with an IQR of 0.543-0.676 and 95% CI of 0.588-0.624. This was significantly outperformed by adding either one or two additional molecular data modalities (Fig. 3a,b; Supp. Fig. S10b, S14), i.e. median AUROC of 0.678 with an IQR of 0.612-0.752 and 95% CI of 0.652-0.692 and AUROC of 0.700 with IQR of 0.651-0.774 and 95% CI of 0.686-0.722, respectively (Wilcoxon rank sum test, $p = 1.9e-16$ and $p = 2.9e-11$, respectively). In essence, molecular features significantly enhanced DSPN incidence prediction.”

Supplemental Figure S13: Baseline models to predict DSPN incidence. Prediction probabilities during testing of negative samples using a) the prevalent DSPN model trained on clinical data alone at F4, b) baseline incidence model trained only on clinical variables at F4 and incidence label at FF4 and c) the full incidence model trained on clinical + molecular variables at F4 and incidence label at FF4. Cases are samples developing DSPN from F4 to FF4, and controls are ones remaining negative.

- **Predicting DSPN cases 6.3-6.7 years into the future implies that one does not even have access to even the DSPN labels from the future. If this concern can be addressed or the claim of future prediction toned down, I am comfortable with recommending the publication of this work.**

Thanks for this comment. We rigorously cross-validated our models according to state-of-the-art machine learning practice. This ensured that DSPN incident labels remained unknown to the trained model, and ultimately is the basis of generalisability. To highlight this, we expanded the method section:

“Fully multi-modal characterised samples were used for final model training. For prevalent DSPN prediction, this was 285 samples (31 cases and 254 controls), whilst for incident DSPN prediction, it was 242 samples (54 cases and 188 controls). We created 100 stratified splits which leveraged 80% samples for feature integration / training, and the remaining 20% for model testing. We further partitioned the 80% training samples into stratified five folds for cross-validation. The cross-validation performance was used as a criterion for the FFS algorithm to select the optimal model. We never used any test data for neither model training nor tuning of model parameters.”

REVIEWERS' COMMENTS:

Reviewer #2 (Remarks to the Author):

All my comments have been addressed.

Reviewer #4 (Remarks to the Author):

Thank you for inviting me to review this paper, developing a machine learning framework (IMML) for predicting DSPN, incorporating multimodal data types including clinical, genomic and proteomic data. Given the wealth of information available in many health datasets, methods of combining different modalities of data are of growing interest, and this paper makes an important advance.

Given the paper has been reviewed previously, I focus on the responses to the previous reviewers. The authors have satisfactorily addressed the comments from reviewer 2. Reviewer 3 in their first comment highlighted the different training tasks, one for prevalence in F4, and the other for predicting incidence between F4 and FF4. In changes to the manuscript, the authors have clarified this issue and the distinction between the two tasks is sufficiently clear. Reviewer 3's second comment suggested use of the model trained on the F4 data should be used as a baseline to make predictions on the FF4 dataset. The authors have made changes to the section 'Molecular data improves DSPN incidence prediction' and have added supplemental Figure S13c. In conjunction with the clarification of the different tasks I believe they have satisfactorily addressed this point. I was unclear about the concern raised by reviewer 3's final comment on 'Predicting DSPN cases 6.3-6.7 years into the future...', but believe this reflected a concern over whether future labels were used in the training – in the rebuttal and revisions, the authors indicate this was not the case.

Overall, I think the authors have adequately responded to the earlier reviewer comments. My only additional comments for the authors to consider are below:

1. In Figure S13, what is "Cohen'd"? I presume this is Cohen's d, but a description and interpretation should be provided, e.g., in the figure caption, to guide the reader
2. In Discussion, the authors highlight the advance in incorporating multimodal data sources. Further brief discussion on the clinical implications of using such a model in practice now, rather than as a proof of concept, would be valuable. The AUROC values are relatively low and suggest that its real world application would be limited, without further development and refinement.

Reviewer #5 (Remarks to the Author):

Title: The Interpretable Multimodal Machine Learning (IMML) framework reveals pathological signatures of distal sensorimotor polyneuropathy

Author:

Prof. Menden and colleagues

Pgs 230-234

As recommended, the authors have included a clinical baseline model (s13 a, b) to determine if molecular features are truly drivers of incident DSPN prediction.

Pgs 557-558

In addition, the authors have clarified, as requested, that the test data was set aside and not used for model training or parameter tuning to ensure the generalizability of the test findings, that could be applied to future test data sets.

Reviewer #2 (Remarks to the Author):

All my comments have been addressed.

Reviewer #4 (Remarks to the Author):

- **Thank you for inviting me to review this paper, developing a machine learning framework (IMML) for predicting DSPN, incorporating multimodal data types including clinical, genomic and proteomic data. Given the wealth of information available in many health datasets, methods of combining different modalities of data are of growing interest, and this paper makes an important advance.**

We appreciate the positive evaluation of our work.

- **Given the paper has been reviewed previously, I focus on the responses to the previous reviewers. The authors have satisfactorily addressed the comments from reviewer 2. Reviewer 3 in their first comment highlighted the different training tasks, one for prevalence in F4, and the other for predicting incidence between F4 and FF4. In changes to the manuscript, the authors have clarified this issue and the distinction between the two tasks is sufficiently clear. Reviewer 3's second comment suggested use of the model trained on the F4 data should be used as a baseline to make predictions on the FF4 dataset. The authors have made changes to the section 'Molecular data improves DSPN incidence prediction' and have added supplemental Figure S13c. In conjunction with the clarification of the different tasks I believe they have satisfactorily addressed this point. I was unclear about the concern raised by reviewer 3's final comment on 'Predicting DSPN cases 6.3-6.7 years into the future...', but believe this reflected a concern over whether future labels were used in the training – in the rebuttal and revisions, the authors indicate this was not the case.**

Thanks for assessing our revisions.

- **Overall, I think the authors have adequately responded to the earlier reviewer comments. My only additional comments for the authors to consider are below:
1. In Figure S13, what is "Cohen'd"? I presume this is Cohen's d, but a description and interpretation should be provided, e.g., in the figure caption, to guide the reader**

Thank you for your comment. Yes it was a typo, "Cohen'd" refers to "Cohen's d", which was used in this context as a measure of effect size. We adjusted the figure, and the figure caption as below:

Supplemental Figure S13: Baseline models to predict DSPN incidence.

Prediction probabilities during testing of negative samples using a) the prevalent DSPN model trained on clinical data alone at F4, b) baseline incidence model trained only on clinical variables at F4 and incidence label at FF4 and c) the full incidence model trained on clinical + molecular variables at F4 and incidence label at FF4. Cases are samples developing DSPN from F4 to FF4, and controls are ones remaining negative. *For each comparison, Cohen's d was used as the measure of the difference between groups.*

- **2. In Discussion, the authors highlight the advance in incorporating multimodal data sources. Further brief discussion on the clinical implications of using such a model in practice now, rather than as a proof of concept, would be valuable. The AUROC values are relatively low and suggest that its real world application would be limited, without further development and refinement.**

Thank you for your comment. We adjusted the discussion part to discuss more about the clinical translation aspect of the model:

Strikingly, our findings were observed in blood instead of biopsies containing neuronal cells which would be more tissue-specific for DSPN but are not accessible in large epidemiological cohort studies. Results of this study highlighted the utility of a less invasive blood-based assay to study complex diseases such as DSPN in clinical practice. Although the prediction performance could be improved further by increasing the quantity and quality of data collection and more advanced machine learning development, we believe that using such a model could both be valuable in clinical practice and for the design of future intervention studies. On the one hand, the early identification of people at elevated risk of DSPN could lead to an intensification of (pharmacological and non-pharmacological) risk factor treatment in these people. On the other hand, our model could be used for an enrichment of high-risk individuals in future intervention trials which could reduce required sample size and therefore the costs to assess novel prevention and treatment options. In the long run, our results indicate potentially actionable biomarkers that could be targeted by novel therapy concepts.

Reviewer #5 (Remarks to the Author):

Title: The Interpretable Multimodal Machine Learning (IMML) framework reveals pathological signatures of distal sensorimotor polyneuropathy

Author:

Prof. Menden and colleagues

Pgs 230-234

As recommended, the authors have included a clinical baseline model (s13 a, b) to determine if molecular features are truly drivers of incident DSPN prediction.

Pgs 557-558

In addition, the authors have clarified, as requested, that the test data was set aside and not used for model training or parameter tuning to ensure the generalizability of the test findings, that could be applied to future test data sets.

Thanks for assessing our manuscript.